# Heat stress-induced activation of MAPK pathway attenuates Atf1-dependent epigenetic inheritance of heterochromatin in fission yeast

Li Sun[†], Libo Liu[†], Chunlin Song[†], Yamei Wang*, Quan-wen Jin*

State Key Laboratory of Cellular Stress Biology, School of Life Sciences, Faculty of Medicine and Life Sciences, Xiamen University, Xiamen, China

**Abstract** Eukaryotic cells are constantly exposed to various environmental stimuli. It remains largely unexplored how environmental cues bring about epigenetic fluctuations and affect heterochromatin stability. In the fission yeast *Schizosaccharomyces pombe*, heterochromatic silencing is quite stable at pericentromeres but unstable at the mating-type (*mat*) locus under chronic heat stress, although both loci are within the major constitutive heterochromatin regions. Here, we found that the compromised gene silencing at the *mat* locus at elevated temperature is linked to the phosphorylation status of Atf1, a member of the ATF/CREB superfamily. Constitutive activation of mitogen-activated protein kinase (MAPK) signaling disrupts epigenetic maintenance of heterochromatin at the *mat* locus even under normal temperature. Mechanistically, phosphorylation of Atf1 impairs its interaction with heterochromatin protein Swi6[HP1], resulting in lower site-specific Swi6[HP1] enrichment. Expression of non-phosphorylatable Atf1, tethering Swi6[HP1] to the *mat3M*-flanking site or absence of the anti-silencing factor Epe1 can largely or partially rescue heat stress-induced defective heterochromatic maintenance at the *mat* locus.

*For correspondence:
wangyamei@xmu.edu.cn (YW);
jinquanwen@xmu.edu.cn (Q-wenJ)

†These authors contributed equally to this work

Competing interest: The authors declare that no competing interests exist.

## Editor's evaluation

This fundamental study demonstrates that MAPK signaling under heat stress phosphorylates ATF in fission yeast leading to de-repression of the mating type locus. The compelling work shows the phosphorylation of ATF reduces its interactions with the heterochromatin protein Swi6 causes its loss at the mating type locus. This is an example of how signaling regulates transcription factor:co-repressor interactions to control gene expression.

## Introduction

Eukaryotic genomes contain two types of chromatin, namely euchromatin and heterochromatin, which are characterized according to their structure and compaction state, and the latter is crucial for regulating the gene expression pattern, cell differentiation, and maintaining genomic stability (*Allshire and Madhani, 2018*; *Bloom, 2014*). In the fission yeast *Schizosaccharomyces pombe*, RNAi machinery contributes to the establishment of major constitutive heterochromatin at pericentromeres, telomeres, and the silent mating-type region (*mat* locus) (*Martienssen and Moazed, 2015*; *Volpe et al., 2002*). In general, transcripts from heterochromatic regions, such as pericentromeric repeats, were processed into double strand small interfering RNAs (siRNAs) by RNase Dicer (Dcr1 in fission yeast) (*Colmenares et al., 2007*), then siRNAs were loaded to Argonaute (Ago1) to finally form functional RNAi-induced transcriptional silencing (RITS) complex, only containing single-stranded siRNAs

(*Verdel et al., 2004*). The RITS complex can target nascent noncoding RNAs from heterochromatic regions through the single-stranded guide siRNAs and subsequently recruit the H3K9 methyltransferase Clr4 to establish H3K9me2/3 (*Bayne et al., 2010*; *Hong et al., 2005*), which can be bound by heterochromatin protein Swi6 and Chp2 through the conserved N-terminal chromo-domain (CD) (*Jacobs and Khorasanizadeh, 2002*; *Jacobs et al., 2001*; *Maison and Almouzni, 2004*). Heterochromatin proteins act as a platform to recruit downstream heterochromatin factors, such as the histone deacetylase (HDAC) Clr3 (*Motamedi et al., 2008*; *Sugiyama et al., 2007*), to initiate heterochromatin assembly. Once established, H3K9me2/3 can be firmly inherited independent of the mechanisms of heterochromatin establishment (*Allshire and Madhani, 2018*).

Heterochromatin plays an essential role in epigenetic gene silencing in organisms ranging from yeast to humans. Epigenetic states of heterochromatin can be stably inherited, but they are also reversible, which is true for not only facultative but also constitutive heterochromatin, and it can be influenced by environmental cues and thus evokes phenotypic variations. Eukaryotic cells are constantly exposed to various environmental stimuli, such as changes in osmotic pressure, oxygen, and temperature. Although the possible impact of the environment on epigenetic regulation has attracted considerable interest, it remains largely unknown how environmental cues bring about epigenetic fluctuations. So far, sporadic studies have demonstrated that heat stress is one of the most prevalent environmental stresses that trigger epigenetic alterations, which may negatively affect early embryonic development in mammals (*Sun et al., 2023*) and eye color-controlling gene inactivation during early larval development in *Drosophila* (*Seong et al., 2011*), or serve as thermosensory input to positively control the rate of vernalization of the flowering plants after winter (*Antoniou-Kourounioti et al., 2018*; *Feil and Fraga, 2012*; *Song et al., 2013*).

In fission yeast, two redundant pathways contribute to establishment and maintenance of heterochromatin at the endogenous silent mating-type region. These two mechanisms rely on two major *cis* elements *cenH* and *REIII* acting as nucleation centers to recruit the H3K9 methyltransferase Clr4 via the RNAi machinery and the RNAi-independent ATF/CREB family proteins Atf1/Pcr1, respectively (*Hall et al., 2002*; *Jia et al., 2004a*; *Kim et al., 2004*; *Thon et al., 1999*; *Yamada et al., 2005*). The initial nucleation and subsequent spreading of heterochromatin are further facilitated by Swi6[HP1] and HDACs (including Clr3 and Clr6) (*Jia et al., 2004a*; *Kim et al., 2004*; *Yamada et al., 2005*). As two major stress-responsive transcription factors, it has been shown that Atf1 and Pcr1 are activated and regulated by Sty1, one of the mitogen-activated protein kinases (MAPKs), in response to high temperature, osmotic, oxidative, and a number of other environmental stresses (*Eshaghi et al., 2010*; *Lawrence et al., 2007*; *Reiter et al., 2008*). Thus, it is plausible to assume that Atf1 and Pcr1 have the potential to render the heterochromatin stability at the silent mating-type region to be more resistant to ambient perturbations. However, contrary to this pre-assumption, recent studies demonstrated that the constitutive heterochromatin at centromeres is propagated stably whereas the epigenetic stability at the *mat* locus in vegetatively growing cells is sensitive to being continuously cultured at elevated temperatures (*Greenstein et al., 2018*; *Nickels et al., 2022*; *Oberti et al., 2015*). It has been established that the protein disaggregase Hsp104 is involved in buffering environmentally induced epigenetic variation at centromeres by dissolving cytoplasmic Dcr1 aggregates (*Oberti et al., 2015*). However, the reason for the absence of the buffering effect on heterochromatin at mating-type region under similar environmental stress remains elusive.

In this study, to explore the possible mechanism underlying the lack of epigenetic stability at the mating-type region in fission yeast under high temperature, we systematically performed genetic analyses combined with biochemical characterization. We found that heat stress-induced phosphorylation of Atf1 negatively influences its recruiting capability toward heterochromatin protein Swi6[HP1], and thus it results in defective Atf1-dependent epigenetic maintenance of heterochromatin at the *mat* locus.

## Results
### Gene silencing within constitutive heterochromatin at the *mat* locus is unstable under heat stress
To examine the stability of heterochromatin under environmental stresses, we first tested the robustness of heterochromatic silencing when cells were grown at 37°C, which is above the permissive

temperature for *S. pombe* and causes acute temperature stress. We employed *S. pombe* strains with an *ade6+* reporter gene placed within two major constitutive heterochromatic regions, represented by pericentromere of chromosome I (*otr1R::ade6+*) and the mating-type region of chromosome II (*mat3M::ade6+*) (*Figure 1A*). Cells with repressed *ade6+* within heterochromatic region gave rise to red colonies under limiting adenine conditions and failed to grow on medium without adenine, whereas de-repressed *ade6+* allowed cells to form white colonies or vigorous growth (*Figure 1B*). Consistent with previous study (*Oberti et al., 2015*), *otr1R::ade6+* cells formed almost fully red colonies at all tested temperatures (*Figure 1B and C*). In contrast, *mat3M::ade6+* cells gave rise to variegated colonies or white colonies with low degrees of redness, or even were able to grow on medium without adenine at 37°C (*Figure 1B and C*). Accordingly, the mRNA levels of *ade6+*, as measured by quantitative RT-PCR, increased by sevenfold in *mat3M::ade6+* cells but only twofold in *otr1R::ade6+* cells at 37°C compared to the same cells grown at 30°C (*Figure 1D*). These results are consistent with recent reports that heterochromatin at pericentromere is largely maintained and that at the *mat* locus seems to be unstable under heat stress (*Greenstein et al., 2018*; *Nickels et al., 2022*; *Oberti et al., 2015*).

To confirm our above observations at high temperature, we used yeast strains in which an *ura4+* reporter gene was inserted into either the mating-type region (*mat3M::ura4+*) or the pericentromere (*otr1R::ura4+*) (*Figure 1—figure supplement 1A*). The silencing of the *ura4+* gene was monitored by poor colony formation on medium lacking uracil and vigorous growth on medium containing the counter-selective drug 5-fluoroorotic acid (5-FOA) (*Figure 1—figure supplement 1B*). Upon being grown at 37°C, we noticed obvious de-repression when the *ura4+* was inserted at *IR-R* element-proximal site within the mating-type region (*mat3M::ura4+*), but not at pericentromere (*Figure 1—figure supplement 1B, C*), indicating a mild loss of heterochromatic gene silencing at the *mat* locus under heat stress.

We also examined the mRNA and translated protein levels of a *gfp+* transgene inserted at *mat3M* locus (*mat3M::gfp+*) or pericentromeric repeat region (*imr1R::gfp+*) (*Figure 1—figure supplement 2A*). Consistent with our results in *mat3M::ade6+* cells, both *gfp+* mRNA levels and GFP protein levels were increased significantly in *mat3M::gfp+* cells, but not in *imr1R::gfp+* cells at 37°C (*Figure 1—figure supplement 2B*). We noticed that the GFP levels were actually slightly decreased in *imr1R::gfp+* cells at 37°C (*Figure 1—figure supplement 2B*), which might be due to the instability of the GFP under heat stress as previously reported (*Ogawa et al., 1995*; *Siemering et al., 1996*).

To further examine whether our observed heterochromatic gene silencing defects at *mat3M* under heat stress is coupled to compromised maintenance of heterochromatin, we performed ChIP followed by quantitative PCR (ChIP-qPCR) to monitor the levels of H3K9me2 and H3K9me3, the hallmarks of heterochromatin. Our results showed that when cells were grown at 37°C, H3K9me2 was reduced modestly at all heterochromatic regions, and intriguingly, H3K9me3 enrichment was similarly reduced within *cenH* element-surrounding regions except pericentromeric repeats and the *cenH* region itself (*Figure 1E*), which shares homology to pericentromeric repeats and is required for nucleation of heterochromatin at the *mat* locus. This is consistent with recent finding that H3K9me3, but not H3K9me2, is a more reliable hallmark for heterochromatin (*Cutter DiPiazza et al., 2021*; *Jih et al., 2017*). Together, these results demonstrated that gene silencing and heterochromatin at the *mat* locus is much more sensitive to high temperature than pericentromeric regions in fission yeast.

## Heat stress compromises reestablishment of a stable epigenetic state of heterochromatin at the *mat* locus

One previous study has revealed that the RNAi mechanism is still functional and actively confers the robustness of epigenetic maintenance of heterochromatin at centromere upon heat stress (*Oberti et al., 2015*). At the *mat* locus, RNAi mechanism is similarly required for the nucleation of heterochromatin at the *cenH* element and subsequent spreading across the entire *mat* locus, but it is dispensable for the maintenance of heterochromatin (*Jia et al., 2004a*; *Kim et al., 2004*). Our observation that H3K9me3 was largely maintained within *cenH* at 37°C (*Figure 1E*) prompted us to investigate whether the defective maintenance of heterochromatin at *mat3M* might be masked by RNAi-mediated de novo heterochromatin assembly. Indeed, in accordance with our assumption, the variegated colonies from *mat3M::ade6+* cells at 37°C restored gene silencing rapidly at normal temperature 30°C after being re-plated on medium containing limited adenine (*Figure 2A and B*). However, when this re-plating assay was applied to *dcr1Δ*, one of the RNAi mutants, a considerable proportion of cells still

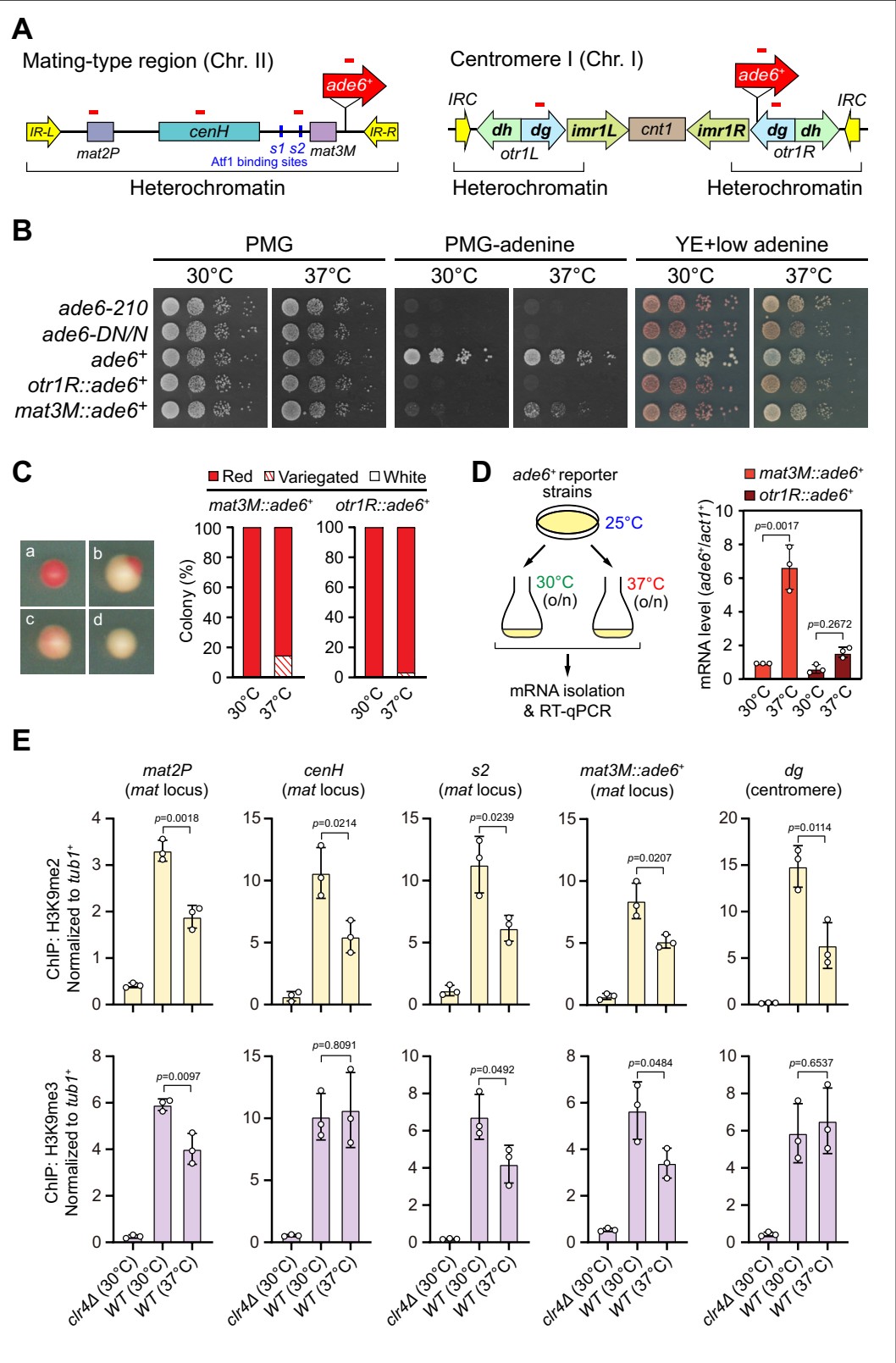

**Figure 1.** Heat stress leads to gene silencing defects at the mating-type region. (**A**) Schematic of an *ade6⁺* reporter gene inserted into mating-type region and pericentromeric region. Primer positions for RT-qPCR or ChIP analysis are indicated (red bars). *cenH*, a DNA element homologous to pericentromeric repeats; *mat2-P* and *mat3-M*, two silent cassettes used for mating-type switching; *IR-L* and *IR-R*, inverted repeats and boundary elements; *s1* and *s2*,

*Figure 1 continued on next page*

*Figure 1 continued*

two Atf1 binding sites; *cnt1*, central core; *imr1*, innermost repeats; *otr1*, outermost repeats; *dg* and *dh*, tandem repeats in *otr*; *IRC*, inverted repeats and boundary elements. (**B**) Expression of the *ade6⁺* reporter monitored by serial dilution spot assay at 30°C and 37°C. The media used were nonselective PMG, selective PMG without adenine, and YE5S with low concentration of adenine. (**C**) Expression of the *ade6⁺* reporter monitored by colony color assay. (Left) Representative colonies of *ade6⁺* reporter fully repressed (red), partially repressed (variegated), and completely expressed (white) on low adenine medium. (Right) Variegated colonies were quantified at 30°C and 37°C. n>500 colonies counted for each sample. (**D**) RT-qPCR analyses of *ade6⁺* reporter. (Left) Schematic depicting the experimental flow of culturing and mRNA extraction. (Right) The relative *ade6⁺* mRNA level was quantified with a ratio between *mat3M::ade6⁺* and *act1⁺* in 30°C samples being set as 1.00. Error bars indicate mean ± standard deviation of three independent experiments. Two-tailed unpaired *t*-test was used to derive p-values. (**E**) ChIP-qPCR analyses of H3K9me2/3 levels at heterochromatic loci. Relative enrichment of H3K9me2/3 was normalized to that of a *tub1⁺* fragment. Error bars represent standard deviation of three experiments. Two-tailed unpaired *t*-test was used to derive p-values.

The online version of this article includes the following source data and figure supplement(s) for figure 1:

**Source data 1.** Raw data of colony color assay, RT-qPCR, H3K9me2/3 ChIP.

**Figure supplement 1.** Heat stress leads to defective silencing of reporter *ura4⁺* at the mating-type region.

**Figure supplement 1—source data 1.** Raw data of RT-qPCR.

**Figure supplement 2.** Heat stress leads to defective silencing of *gfp⁺* reporter gene at the mating-type region.

**Figure supplement 2—source data 1.** Raw data of GFP level measurement, RT-qPCR.

**Figure supplement 2—source data 2.** Full raw unedited blot (mat3M-GFP) for *Figure 1—figure supplement 2B*.

**Figure supplement 2—source data 3.** Full raw unedited blot (Cdc2) for *Figure 1—figure supplement 2B*.

**Figure supplement 2—source data 4.** Full raw unedited blot (imr1R-GFP) for *Figure 1—figure supplement 2B*.

**Figure supplement 2—source data 5.** Full raw unedited blot (Cdc2) for *Figure 1—figure supplement 2B*.

**Figure supplement 2—source data 6.** Uncropped blots for *Figure 1—figure supplement 2*.

emerged as variegated colonies (designated as *dcr1Δᵛ*), which was in sharp contrast to wild type cells (*Figure 2B*). Our RT-qPCR analyses confirmed that the mRNA levels of the reporter *mat3M::ade6⁺* and *cenH* increased similarly and dramatically in *dcr1Δᵛ* cells compared to those in *dcr1Δᴿ* (refers to 'red' colonies) cells at both 30°C and 37°C, whereas *dg* transcription was de-repressed irrespective of red or variegated colonies or temperatures (*Figure 2C*, *Figure 2—figure supplement 1*). Furthermore, the de-repression also correlated with severe reduction in H3K9me3 levels within the entire *mat* locus in *dcr1Δᵛ* cells (*Figure 2D*). These data strongly suggested that heat stress also leads to defective reestablishment of stable heterochromatin at the *mat* locus.

## Phosphorylation of Atf1 causes heat stress-induced defective epigenetic maintenance of the *mat* locus

It has been recently established that a composite DNA element within *REIII* at the *mat* locus contains binding sequences for Atf1/Pcr1, Deb1, and the origin recognition complex (ORC), which act together in epigenetic maintenance of heterochromatin in the absence of RNAi nucleation with Atf1 as the dominating contributor (*Wang et al., 2021*). Two previous studies showed that extracellular stresses induce phosphorylation of *Drosophila* dATF-2 and mouse ATF7, two homologs of *S. pombe* Atf1, this provokes their release from heterochromatin and thus disrupts heterochromatic maintenance (*Liu et al., 2019*; *Seong et al., 2011*). Given that Atf1 can be phosphorylated by MAPK under heat stress (*Samejima et al., 1997*; *Shiozaki et al., 1998*), we surmised that heat stress-induced phosphorylation of Atf1 might similarly cause its release from the *mat* locus in fission yeast. Surprisingly, ChIP-qPCR analyses showed that Atf1 abundance at the *mat* locus was not altered at 37°C, although it was indeed increased modestly at *SPCC320.03*, an euchromatic target of Atf1/Pcr1 (*Eshaghi et al., 2010*; *Figure 3A*).

Atf1 contains 11 putative MAPK phosphorylation sites, 10 out of them are within the first half of Atf1 (*Lawrence et al., 2007*; *Figure 3B*). Thus, we asked whether the phosphorylation status of Atf1 is causative to defective maintenance of heterochromatin at the *mat* locus under heat stress. To test this, we constructed strains carrying ectopically expressed *HA-atf1*, *HA-atf1(10A/I)*, or *HA-atf1(10D/E)* under the control of the *sty1⁺* promoter ($P_{sty1}$) with the endogenous *atf1⁺* gene deleted (*Salat-Canela*

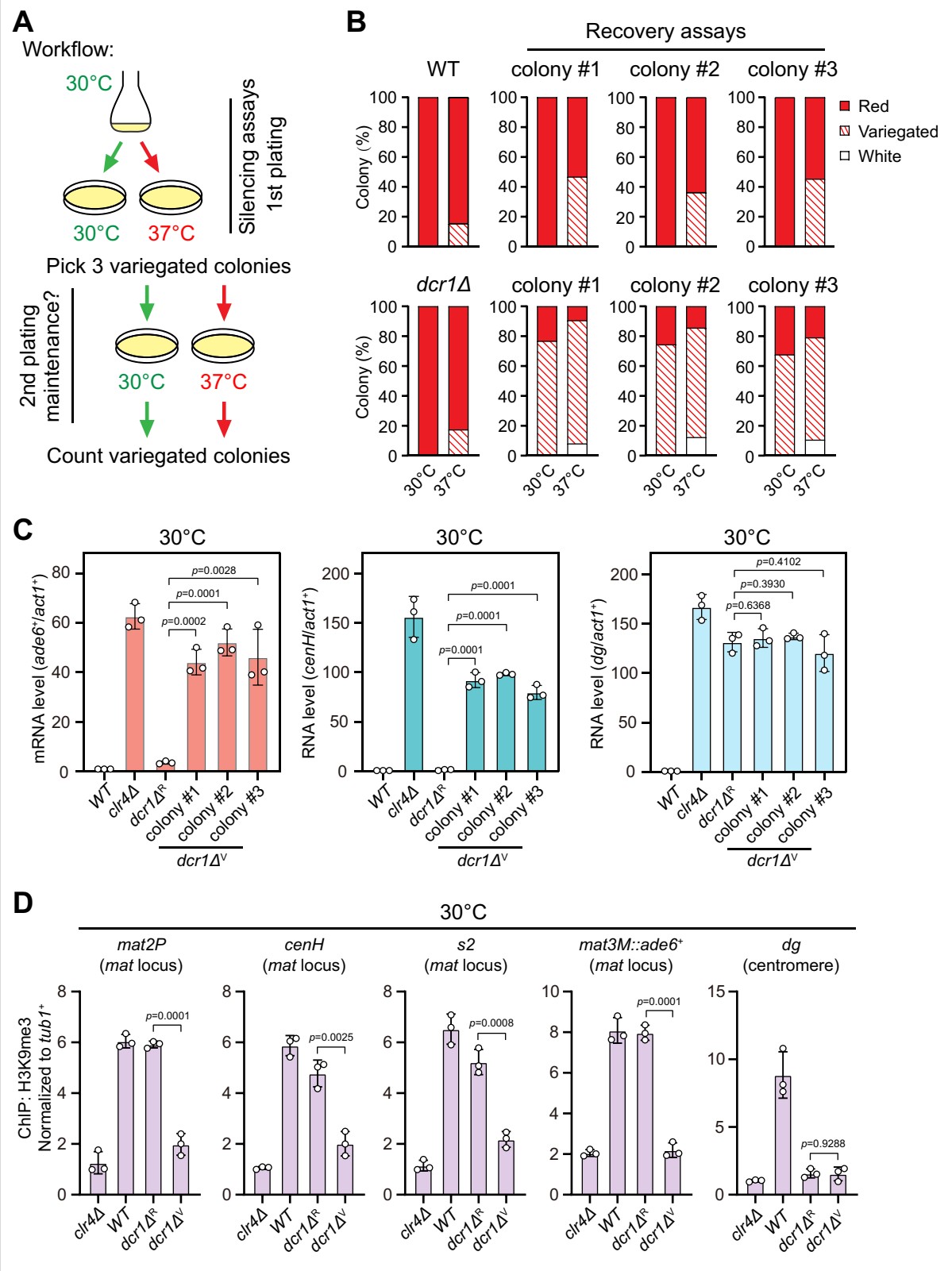

**Figure 2.** Heat stress compromises reestablishment of a stable epigenetic state of heterochromatin at the mating-type region. (**A**) Workflow of gene silencing recovery assays. First plating: strains were plated on low adenine medium at 30°C and 37°C. Second plating: three variegated colonies (*mat3M::ade6+* was partially repressed) from low adenine plates at 37°C were collected, resuspended in water, and then directly re-plated on low adenine medium and grown at 30°C or 37°C. Variegated colonies were counted to assess the gene silencing defect. (**B**) Quantified results of

*Figure 2 continued on next page*

*Figure 2 continued*

*mat3M::ade6⁺* gene silencing recovery assays. Variegated colonies on low adenine medium from first plating and second plating were counted. n>500 colonies counted for each sample. (**C**) RT-qPCR analyses of *mat3M::ade6⁺* reporter, *cenH* and *dg* transcripts. *dcr1Δ*$^R$ and *dcr1Δ*$^V$ denote red colonies and variegated colonies respectively when *dcr1Δ* cells were grown on low adenine plate at 37°C. Three variegated colonies were picked and re-plated on low adenine medium and grown at 30°C. The relative transcript level was quantified with a ratio between respective transcript and *act1⁺* in 30°C wild type samples being set as 1.00. Error bars indicate mean ± standard deviation of three independent experiments. Two-tailed unpaired *t*-test was used to derive p-values. (**D**) ChIP-qPCR analyses of H3K9me3 levels at heterochromatic loci. Samples were collected as in (**C**). Relative enrichment of H3K9me3 was normalized to that of a *tub1⁺* fragment. Error bars represent standard deviation of three experiments. Two-tailed unpaired *t*-test was used to derive p-values.

The online version of this article includes the following source data and figure supplement(s) for figure 2:

**Source data 1.** Raw data of colony color assay, RT-qPCR, H3K9me3 ChIP.

**Figure supplement 1.** Comparison of expression of *mat3M::ade6⁺* reporter, *cenH* and *dg* in *dcr1Δ* background at 30°C and 37°C after heat stress.

**Figure supplement 1—source data 1.** Raw data of RT-qPCR.

---

*et al., 2017*). Alleles of *HA-atf1(10A/I)* and *HA-atf1(10D/E)* harbor 10 non-phosphorylatable alanines (Ala, A) and isoleucines (Ile, I), or phosphomimetic aspartic acids (Asp, D) and glutamic acids (Glu, E) replacing serines or threonines, respectively (*Salat-Canela et al., 2017*). Our immunoblotting analyses showed that the protein levels of HA-Atf1 and HA-Atf1*(10A/I)* were comparable (*Figure 3C*). Intriguingly, cells expressing P$_{sty1}$-*HA-atf1(10A/I)* fully rescued the gene silencing defects observed in P$_{sty1}$-*HA-atf1* cells at 37°C, which was confirmed by RT-qPCR analyses (*Figure 3D and E*). Consistently, more Atf1 was maintained at the *mat* locus in P$_{sty1}$-*HA-atf1(10A/I)* cells compared to that in P$_{sty1}$-*HA-atf1* cells grown at both 30°C and 37°C (*Figure 3F*). Moreover, ChIP-qPCR analyses showed that the enrichment of H3K9me3 and heterochromatin protein Swi6$^{HP1}$ at the *mat* locus in P$_{sty1}$-*HA-atf1(10A/I)* cells was also restored to the level of wild type cells (*Figure 3G*). To our surprise, cells expressing P$_{sty1}$-*HA-atf1(10D/E)* were almost completely unable to grow at 37°C (*Figure 3—figure supplement 1*), which may be due to the toxicity caused by constitutive level of Atf1(10D/E).

Next, in order to validate our above observation that the phosphorylation status of physiological levels of Atf1 affects heterochromatin maintenance at the *mat* locus, we constructed Atf1 phosphorylation mutant strains with Atf1(10A/I) or Atf1(10D/E) expressed under the control of the endogenous *atf1* promoter (P$_{atf1}$). We noticed that P$_{atf1}$-*atf1(10D/E)* allowed cells carrying *mat3M::ade6⁺* to be viable and form variegated or white colonies at 37°C, whereas P$_{atf1}$-*atf1(10A/I) mat3M::ade6⁺* cells formed red colonies with higher degrees of redness than P$_{atf1}$-*atf1(WT)* and P$_{atf1}$-*atf1(10D/E)* cells (*Figure 3—figure supplement 2A, B*). More strikingly, P$_{atf1}$-*atf1(10D/E)* also visibly compromised epigenetic silencing even at 30°C (*Figure 3—figure supplement 2A, B*). Although protein levels of Atf1(10A/I) and Atf1(10D/E) driven by endogenous *atf1* promoter were apparently lower than those expressed from *sty1* promoter (*Figure 3—figure supplement 2C*), the mRNA levels of *mat3M::ade6⁺* reporter measured by RT-qPCR were still reduced in P$_{atf1}$-*atf1(10A/I)* cells and elevated in P$_{atf1}$-*atf1(10D/E)* cells compared to wild type cells (*Figure 3—figure supplement 2D*), which was not accompanied with largely altered binding of Atf1 at the *mat* locus (*Figure 3—figure supplement 2E*).

Taken together, these data demonstrated that phosphorylation of Atf1 causes heat stress-induced defective epigenetic maintenance at the *mat* locus.

## Heat stress-induced phosphorylation of Atf1 reduces Atf1 but not Pcr1 binding affinity to Swi6$^{HP1}$

Previous studies have shown that Atf1 contributes to the epigenetic maintenance of the *mat* locus by actively recruiting the H3K9 methyltransferase Clr4, heterochromatin protein Swi6, and two HDACs Clr3 and Clr6 (*Jia et al., 2004a*; *Kim et al., 2004*). To explore whether phosphorylation of Atf1 compromises its capability of recruiting these heterochromatic factors at the *mat* locus under heat stress, we performed in vitro pull-down assays using yeast lysates prepared from cultures grown at either 30°C or 37°C and bacterially expressed GST-Clr3, GST-Clr4, MBP-Clr6, and His-Swi6. We found that Atf1 from cells grown at 37°C almost completely lost its binding to Swi6$^{HP1}$ but not the other three heterochromatic proteins (*Figure 4A* and *Figure 4—figure supplement 1A*). Very interestingly, non-phosphorylatable Atf1(10A/I) from 37°C cultures remained binding to Swi6$^{HP1}$ more efficiently than

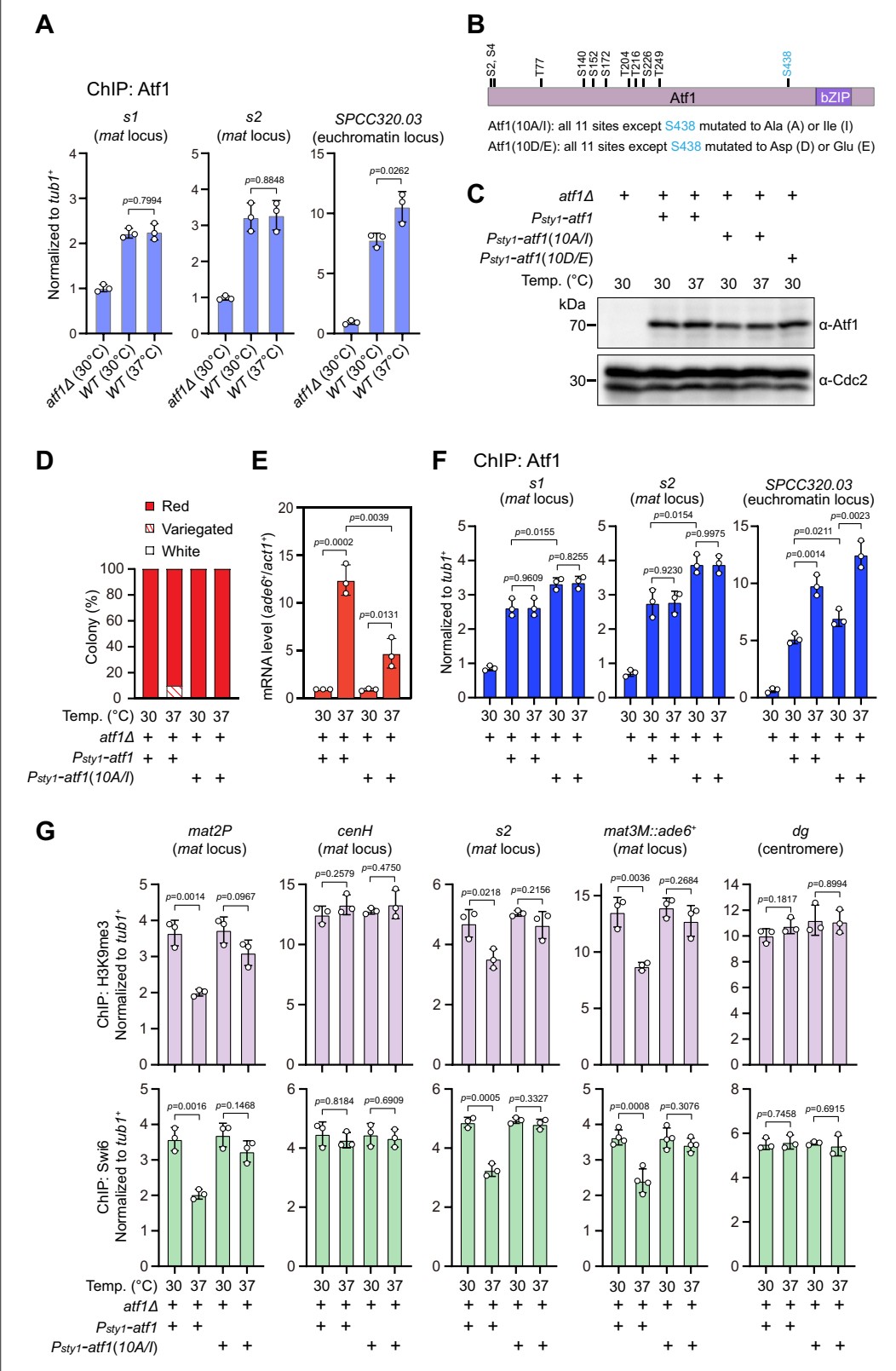

**Figure 3.** Heat stress-induced defective heterochromatic maintenance at the mating-type region can be rescued by non-phosphorylatable Atf1(10A/I). (**A**) ChIP-qPCR analyses of Atf1 levels at two Atf1 binding sites (*s1*, *s2*) within mating-type region and an euchromatic target of Atf1 (SPCC320.03). Relative enrichment of Atf1 is normalized to that of a *tub1⁺* fragment. Error bars represent standard deviation of three experiments. Two-tailed unpaired *t*-test

*Figure 3 continued*

was used to derive p-values. (**B**) Schematic depiction of the Atf1 protein with the substitutions of the 10 putative phosphorylation sites to alanines or isoleucines (10A/I), or aspartic acids or glutamic acids (10D/E) indicated. (**C**) Western blotting analyses of the protein level of Atf1 in *atf1Δ* background cells expressing HA-Atf1, HA-Atf1(10A/I), or HA-Atf1(10D/E) under the control of *sty1+* promoter. (**D**) Expression of the *mat3M::ade6+* reporter monitored by colony color assay in *atf1Δ* cells expressing $P_{sty1}$-HA-atf1 or $P_{sty1}$-HA-atf1(10A/I) as in **Figure 1C**. n>500 colonies counted for each sample. (**E**) RT-qPCR analyses of the *mat3M::ade6+* reporter in *atf1Δ* cells expressing $P_{sty1}$-HA-atf1 or $P_{sty1}$-HA-atf1(10A/I). (**F**) ChIP-qPCR analyses of Atf1 levels at two Atf1 binding sites within mating-type region and an euchromatic target of Atf1 (SPCC320.03) in *atf1Δ* cells expressing $P_{sty1}$-HA-atf1 or $P_{sty1}$-HA-atf1(10A/I). (**G**) ChIP-qPCR analyses of H3K9me3 and Swi6 levels at heterochromatic loci in *atf1Δ* cells expressing $P_{sty1}$-HA-atf1 or $P_{sty1}$-HA-atf1(10A/I).

The online version of this article includes the following source data and figure supplement(s) for figure 3:

**Source data 1.** Raw data of colony color assay, RT-qPCR, Atf1/Swi6/H3K9me3 ChIP.

**Source data 2.** Full raw unedited blot (Atf1) for **Figure 3C**.

**Source data 3.** Full raw unedited blot (Cdc2) for **Figure 3C**.

**Source data 4.** Uncropped blots for **Figure 3C**.

**Figure supplement 1.** Expression of Atf1(10D/E) under the control of *sty1* promoter ($P_{sty1}$) leads to lethality at 37°C.

**Figure supplement 2.** Atf1 phosphorylation mutants Atf1(10A/I) and Atf1(10D/E) expressed under the endogeneous *atf1* promoter enhance or reduce *mat3M::ade6+* silencing respectively.

**Figure supplement 2—source data 1.** Full raw unedited blots (Atf1 and Cdc2) for **Figure 3—figure supplement 2C**.

**Figure supplement 2—source data 2.** Uncropped blots for **Figure 3—figure supplement 2**.

**Figure supplement 2—source data 3.** Raw data of RT-qPCR, Atf1 ChIP.

---

wild type Atf1, and phosphomimetic Atf1(10D/E) rendered weak binding to Swi6[HP1] even when it was derived from 30°C cultures (**Figure 4A**).

Our ChIP-qPCR analyses confirmed that Swi6[HP1] enrichment was indeed reduced at the *mat* locus, but not at pericentromeric repeats under heat stress (**Figure 4B**). In addition, we also noticed that Clr3 level was slightly decreased at regions distal to the *cenH* nucleation center (i.e. *mat2P* and *mat3M*) but not at pericentromeres and the *cenH* under heat stress (**Figure 4—figure supplement 1B**). This was in sharp contrast to the actually slight increase of the binding between Clr3 and Atf1 at 37°C detected by in vitro pull-down assays (**Figure 4—figure supplement 1A**). Consistent with our in vitro pull-down assays, Clr4 and Clr6 enrichment was not altered at all tested heterochromatin sites under heat stress (**Figure 4—figure supplement 1C, D**).

It has been shown that Atf1 and Pcr1 can form heterodimer and both are activated by MAPK in response to a variety of environmental stresses (**Eshaghi et al., 2010**; **Lawrence et al., 2007**; **Reiter et al., 2008**). Also, both Atf1 and Pcr1 bind to the *cis* elements flanking the *mat3M* cassette and interact with Swi6[HP1] to facilitate epigenetic inheritance of heterochromatin (**Jia et al., 2004a**; **Kim et al., 2004**; **Wang et al., 2021**). We wondered whether binding of Pcr1 to Swi6[HP1] was also compromised at high temperature. To test this, we performed in vitro pull-down assays using yeast lysates prepared from *atf1Δ*, $P_{sty1}$-HA-atf1(WT), $P_{sty1}$-HA-atf1(10A/I), or $P_{sty1}$-HA-atf1(10D/E) strains carrying *pcr1-3xFlag*. In contrary to Atf1, the binding of Pcr1 to Swi6[HP1] was not affected by either heat stress or phosphorylation status of Atf1 (**Figure 4—figure supplement 2A**). In addition, absence of *atf1* or Atf1 phospho mutants did not alter enrichment of Pcr1 at the *mat* locus (**Figure 4—figure supplement 2B**), though $P_{sty1}$-HA-atf1(10A/I) elevated protein abundance of Pcr1 at both 30°C and 37°C (**Figure 4—figure supplement 2A**). These data also indicated that Atf1 and Pcr1 respond differently and separately to heat stress in heterochromatin maintenance at the *mat* locus.

Together, all these results suggested that most likely phosphorylation of Atf1 induced by heat stress disrupts Swi6[HP1] binding affinity to Atf1 but not to Pcr1, and thus attenuates Swi6[HP1] abundance at the *mat* locus, this consequently causes the defective maintenance of heterochromatin specifically at this site.

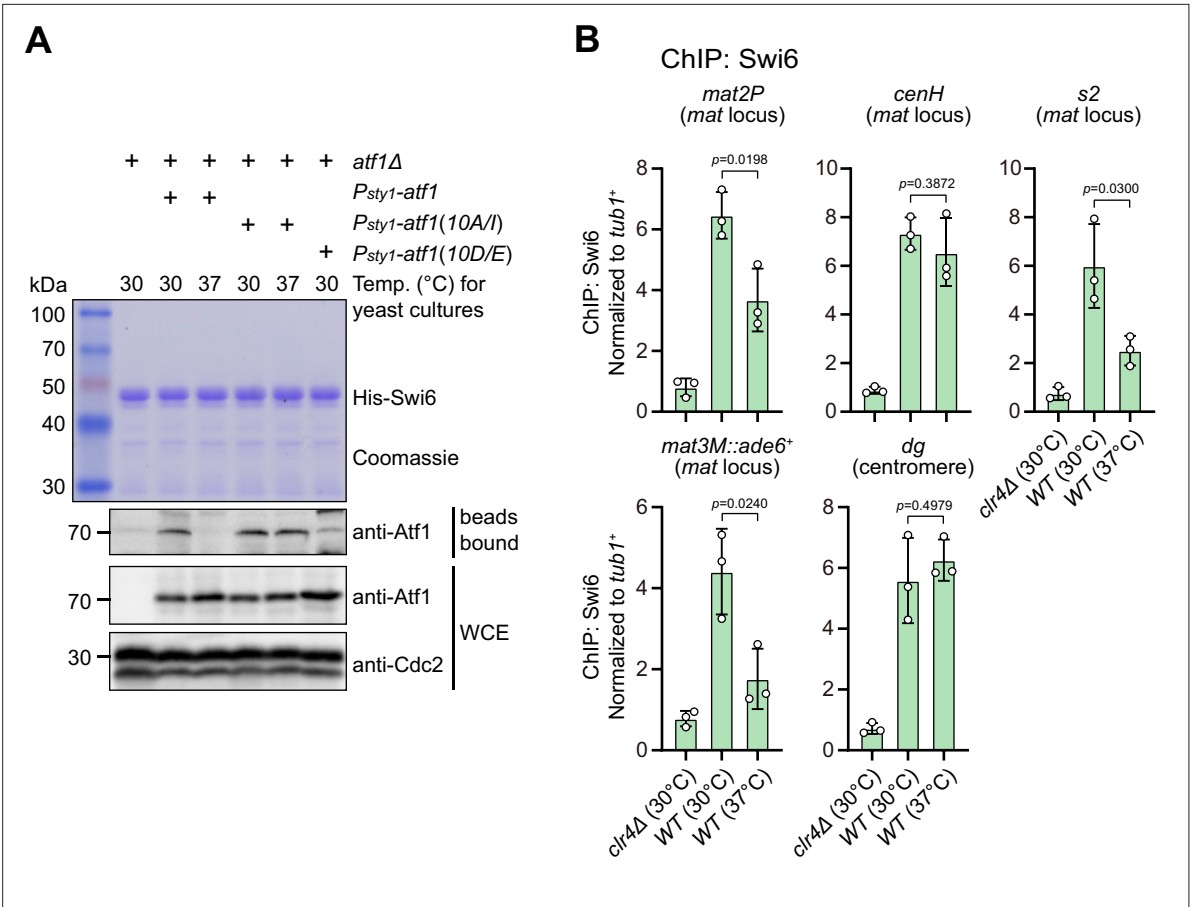

**Figure 4.** Phosphorylation of Atf1 impairs its interaction with Swi6[HP1]. (**A**) Binding affinity between Atf1 and Swi6[HP1] is maintained for non-phosphorylatable Atf1(10A/I) at 37°C, but disrupted for phosphomimetic Atf1(10D/E) even at 30°C. Yeast lysates from *atf1Δ* cells expressing $P_{sty1}$-HA-*atf1*, $P_{sty1}$-HA-*atf1(10A/I)*, or $P_{sty1}$-HA-*atf1(10D/E)* grown at either 30°C or 37°C were incubated with bacteria-expressed 6His-Swi6 in in vitro pull-down assays. Bound and total Atf1 were detected by immunoblotting with Cdc2 used as a loading control. Results are representative of three independent experiments. (**B**) ChIP-qPCR analyses of Swi6 levels at heterochromatic loci. Relative enrichment of Swi6 was normalized to that of a $tub1^+$ fragment. Error bars represent standard deviation of three experiments. Two-tailed unpaired *t*-test was used to derive p-values.

The online version of this article includes the following source data and figure supplement(s) for figure 4:

**Source data 1.** Raw data of Swi6 ChIP.

**Source data 2.** Full raw unedited Coomassie gel (His-Swi6) for **Figure 4A**.

**Source data 3.** Full raw unedited blot (bead bound-Atf1) for **Figure 4A**.

**Source data 4.** Full raw unedited blot (WCE-Atf1) for **Figure 4A**.

**Source data 5.** Full raw unedited blot (WCE-Cdc2) for **Figure 4A**.

**Source data 6.** Uncropped blots for **Figure 4A**.

**Figure supplement 1.** In vitro binding assay of association between Clr3, Clr4, or Clr6 and Atf1, and ChIP-qPCR analyses of their enrichment at different heterochromatic regions under heat stress.

**Figure supplement 1—source data 1.** Raw data of in vitro binding assay, Clr3/Clr4/Clr6 ChIP.

**Figure supplement 1—source data 2.** Full raw unedited Coomassie gel for **Figure 4—figure supplement 1A**.

**Figure supplement 1—source data 3.** Full raw unedited blot (bead bound-Atf1) for **Figure 4—figure supplement 1A**.

**Figure supplement 1—source data 4.** Full raw unedited blot (WCE-Atf1) for **Figure 4—figure supplement 1A**.

**Figure supplement 1—source data 5.** Full raw unedited blot (WCE-Cdc2) for **Figure 4—figure supplement 1A**.

**Figure supplement 1—source data 6.** Uncropped blots for **Figure 4—figure supplement 1A**.

**Figure supplement 2.** Phosphorylation status of Atf1 does not impair interaction between Pcr1 and Swi6[HP1] and alter Pcr1 binding within *mat* locus.

**Figure supplement 2—source data 1.** Full raw unedited gel (Coomassie) for **Figure 4—figure supplement 2A**.

*Figure 4 continued on next page*

*Figure 4 continued*

**Figure supplement 2—source data 2.** Full raw unedited blot (bead bound-Pcr1x3Flag) for *Figure 4—figure supplement 2A*.

**Figure supplement 2—source data 3.** Full raw unedited blot (WCE-Pcr1x3Flag) for *Figure 4—figure supplement 2A*.

**Figure supplement 2—source data 4.** Full raw unedited blot (WCE-Cdc2) for *Figure 4—figure supplement 2A*.

**Figure supplement 2—source data 5.** Uncropped gel and blots for *Figure 4—figure supplement 2A*.

**Figure supplement 2—source data 6.** Raw data of Pcr1-3xFlag ChIP.

## Constitutive activation of MAPK signaling leads to Sty1 kinase-dependent defective epigenetic maintenance of heterochromatin at the *mat* locus under normal temperature

It has been well established that the MAPK Sty1 is constitutively activated in *wis1-DD* mutant (carrying S469D and T473D mutations), and Sty1 phosphorylates and activates Atf1 in response to high temperature (*Eshaghi et al., 2010*; *Lawrence et al., 2007*; *Reiter et al., 2008*). It is fairly possible that permanent activation of Sty1 may also lead to defective epigenetic maintenance of heterochromatin at the *mat* locus even at 25°C or 30°C. To test this possibility, we employed a yeast strain in which *cenH* site was replaced with an *ade6⁺* reporter gene (*kΔ::ade6⁺*) (*Grewal and Klar, 1996*; *Thon and Friis, 1997*) to remove RNAi-mediated heterochromatin nucleation (*Figure 5A*), and *kΔ::ade6⁺* displays one of two distinct statuses: being expressed (*ade6*-on) or being silenced (*ade6*-off). In *ade6*-off cells, Atf1 becomes the major determinant factor for epigenetic maintenance of heterochromatin at the *mat* locus (*Wang and Moazed, 2017*; *Wang et al., 2021*). Notably, cells expressing two copies, but not one copy, of *wis1-DD* showed severe gene silencing defects (*Figure 5B* and *Figure 5—figure supplement 1B*), and consistently the mRNA levels of the *kΔ::ade6⁺* increased dramatically in these cells (*Figure 5C* and *Figure 5—figure supplement 1C*). Our immunoblotting results confirmed that the levels of both the active form of Sty1 (i.e. phosphorylated Sty1) and its downstream effector Atf1 increased in *wis1-DD* mutants regardless of its copy number (*Figure 5D* and *Figure 5—figure supplement 1D*), indicating constitutive activation of the Wis1/Sty1-mediated MAPK signaling. Furthermore, in vitro pull-down assays demonstrated that the interaction between Atf1 and Swi6ᴴᴾ¹ was also largely disrupted in *wis1-DD* mutants (*Figure 5E* and *Figure 5—figure supplement 1E*). Consistently, ChIP-qPCR analyses showed that the abundance of both H3K9me3 and Swi6ᴴᴾ¹ bound at the *mat* locus but not at pericentromere decreased dramatically in cells with two copies of *wis1-DD* (*Figure 5F* and *Figure 5—figure supplement 1F*).

Intriguingly, removal of Sty1 kinase activity by introducing either *sty1* deletion mutant (*sty1Δ*) or ATP analogue-sensitive mutant of *sty1* (*sty1-T97A*, i.e. *sty1-as2*) (*Zuin et al., 2010*) into *wis1-DD* mutant background could relieve the negative effect of constitutive activation of MAPK Sty1 on *kΔ::ade6⁺* reporter gene silencing, binding affinity between Atf1 and Swi6ᴴᴾ¹ and heterochromatin stability at the *mat* locus (*Figure 5* and *Figure 5—figure supplement 2*). Therefore, our data lent support to the idea that constitutive activation of MAPK signaling and resulted Atf1 phosphorylation can also eventually lead to defective epigenetic maintenance and inheritance of heterochromatin at the *mat* locus under normal temperatures.

## Identification of major Sty1-dependent phosphorylation sites in Atf1 upon heat stress

Originally, a total of 11 serine or threonine residues of Atf1 were considered to be putative MAPK phosphorylation sites solely based on their fit with MAPK phosphorylation consensus S/P or T/P (*Lawrence et al., 2007*; *Figure 3B*). However, one subsequent study demonstrated that mutating only the central six serine or threonine sites (S152, S172, T204, T216, S226, and T249) within Atf1 to aspartic or glutamic acid (i.e. Atf1(6D/E)) to mimic phosphorylation is as sufficient as Atf1(11D/E) and Atf1(10D/E) mutants for transcriptional activation and oxidative stress survival (*Salat-Canela et al., 2017*). It has also been recently shown that, similar to Atf1(11D/E) mutant, deletion of the central domain (named as 6P domain) in Atf1 harboring six serine or threonine sites can efficiently block heterochromatin assembly capacity of Atf1 at the *mat* locus (*Fraile et al., 2022*). All these previous observations raised a possibility that only one or a few Sty1-dependent phosphorylation sites within Atf1 may play the major role in regulating epigenetic maintenance of heterochromatin at the mating-type region.

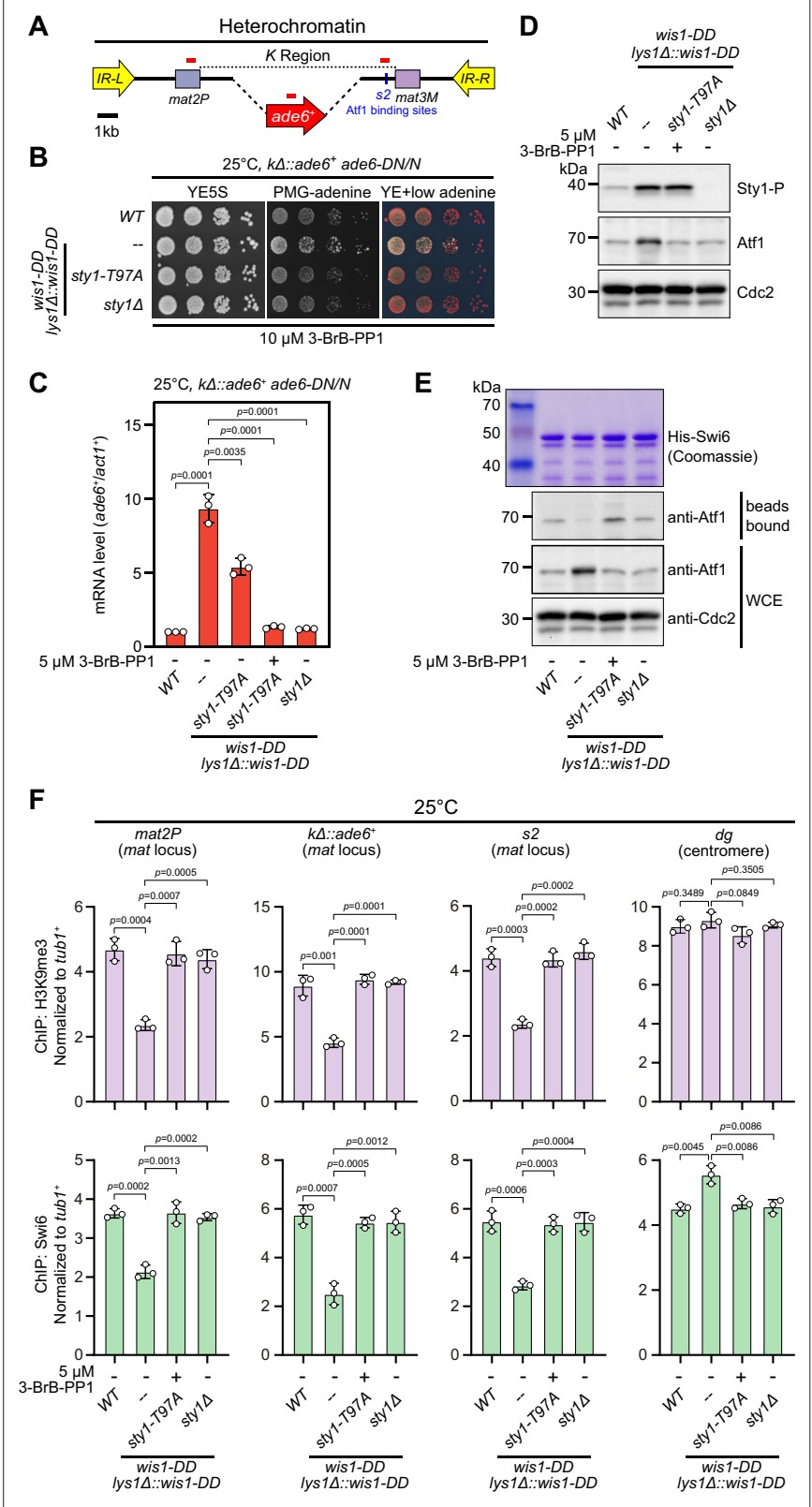

**Figure 5.** Constitutive activation of mitogen-activated protein kinase (MAPK) signaling pathway leads to Sty1 kinase-dependent defective epigenetic maintenance of heterochromatin at the mating-type region. (**A**) Schematic of the mating-type region in $k\Delta::ade6^+$ strain. A 7.5 kb DNA sequence (*K* region) between *mat2P* and *mat3M* locus was replaced with $ade6^+$ reporter. Primer positions for RT-qPCR or ChIP analysis are indicated (red

*Figure 5 continued on next page*

*Figure 5 continued*

bars). (**B**) Expression of the *kΔ::ade6⁺* reporter monitored by serial dilution spot assay at 25°C as in *Figure 1B*. Constitutive activation of one of the MAPK signaling pathways was achieved by expressing *wis1-DD* (*wis1-S469D;T473D*) mutant at endogenous locus and ectopically at *lys1⁺* (*lys1Δ::wis1-DD*) simultaneously. Note that plates were incubated at 25°C because *sty1Δ* mutant is temperature-sensitive. (**C**) RT-qPCR analyses of the *kΔ::ade6⁺* reporter. (**D**) Western blotting analyses of the phosphorylated Sty1 (Sty1-P) and the total protein of Atf1. (**E**) Binding affinity between Atf1 and Swi6^HP1 was detected by in vitro pull-down assays as in *Figure 4A*. Yeast lysates were prepared from wild type or *wis1-DD* cells grown at 25°C. Results are representative of three independent experiments. (**F**) ChIP-qPCR analyses of H3K9me3 and Swi6 levels at heterochromatic loci in wild type and *wis1-DD* cells grown at 25°C. Note that *sty1-T97A* was inhibited with 5 μM or 10 μM 3-BrB-PP1 when cells were grown in liquid cultures or plates respectively.

The online version of this article includes the following source data and figure supplement(s) for figure 5:

**Source data 1.** Raw data of RT-qPCR, Swi6/H3K9me3 ChIP.

**Source data 2.** Full raw unedited blots (Sty1-P, Atf1, and Cdc2) for *Figure 5D*.

**Source data 3.** Uncropped blots for *Figure 5D*.

**Source data 4.** Full raw unedited Coomassie gel for *Figure 5E*.

**Source data 5.** Full raw unedited blot (bead bound-Atf1) for *Figure 5E*.

**Source data 6.** Full raw unedited blot (WCE-Atf1) for *Figure 5E*.

**Source data 7.** Full raw unedited blot (WCE-Cdc2) for *Figure 5E*.

**Source data 8.** Uncropped gel and blots for *Figure 5E*.

**Figure supplement 1.** Constitutive activation of mitogen-activated protein kinase (MAPK) signaling pathway leads to defective epigenetic maintenance of heterochromatin at the mating-type region.

**Figure supplement 1—source data 1.** Raw data of RT-qPCR, Swi6/H3K9me3 ChIP.

**Figure supplement 1—source data 2.** Full raw unedited blot (Sty1-P) for *Figure 5—figure supplement 1D*.

**Figure supplement 1—source data 3.** Full raw unedited blot (Atf1) for *Figure 5—figure supplement 1D*.

**Figure supplement 1—source data 4.** Full raw unedited blot (Cdc2) for *Figure 5—figure supplement 1D*.

**Figure supplement 1—source data 5.** Uncropped blots for *Figure 5—figure supplement 1D*.

**Figure supplement 1—source data 6.** Full raw unedited gel (Coomassie) for *Figure 5—figure supplement 1E*.

**Figure supplement 1—source data 7.** Full raw unedited blot (beads bound-Atf1) for *Figure 5—figure supplement 1E*.

**Figure supplement 1—source data 8.** Full raw unedited blot (WCE-Atf1 and Cdc2) for *Figure 5—figure supplement 1E*.

**Figure supplement 1—source data 9.** Uncropped gel and blots for *Figure 5—figure supplement 1E*.

**Figure supplement 2.** Serial dilution spot assay of expression of the *kΔ::ade6⁺* reporter in *wis1-DD sty1Δ* and *wis1-DD sty1-T97A* mutants.

To more exactly identify residues in Atf1 which undergo Sty1-dependent phosphorylation in vivo under heat stress, we set out to purify HA-Atf1 from both wild type and *sty1Δ* cells being incubated at 30°C or 37°C for 5 hr (*Figure 6A*). Subsequent mass spectrometry analyses revealed that phosphorylation of at least six residues (T77, S115, S166, S172, T204, and T249) and three other residues (S140, S152, and S226) within the central portion of Atf1 were either specifically dependent or independent on the presence of Sty1 at 37°C (*Figure 6A*, *Figure 6—figure supplement 1* and *Figure 6—figure supplement 2*). Interestingly, we found that mutations of all six sites to either non-phosphorylatable alanines (Ala, A) and isoleucines (Ile, I) (i.e. Atf1(6A/I)), or phosphomimetic aspartic acids (Asp, D) and glutamic acids (Glu, E) (i.e. Atf1(6D/E)) decreased or increased the silencing and mRNA levels of *mat3M::ade6⁺* reporter respectively, although the effect was not as strong as in Atf1(10A/I) or Atf1(10D/E) mutants (*Figure 6B and C*). We also noticed that mutating four out of six these sites led to only modest effect on expression of *mat3M::ade6⁺* at high temperature (*Figure 6B and C*).

We noticed that three residues Ser2, Ser4, and Ser438, which are among the 11 MAPK phosphorylation consensus sites at the most N-terminus of Atf1 or next to the bZIP domain respectively, were not identified as Sty1-dependent phosphorylation site in our mass spectrometry analyses (*Figure 6A*). This was most likely due to the failed recovery of the Ser2-, Ser4-, and Ser438-containing peptides (see *Figure 6—figure supplement 1*) derived from technical limitations of mass spectrometry. We

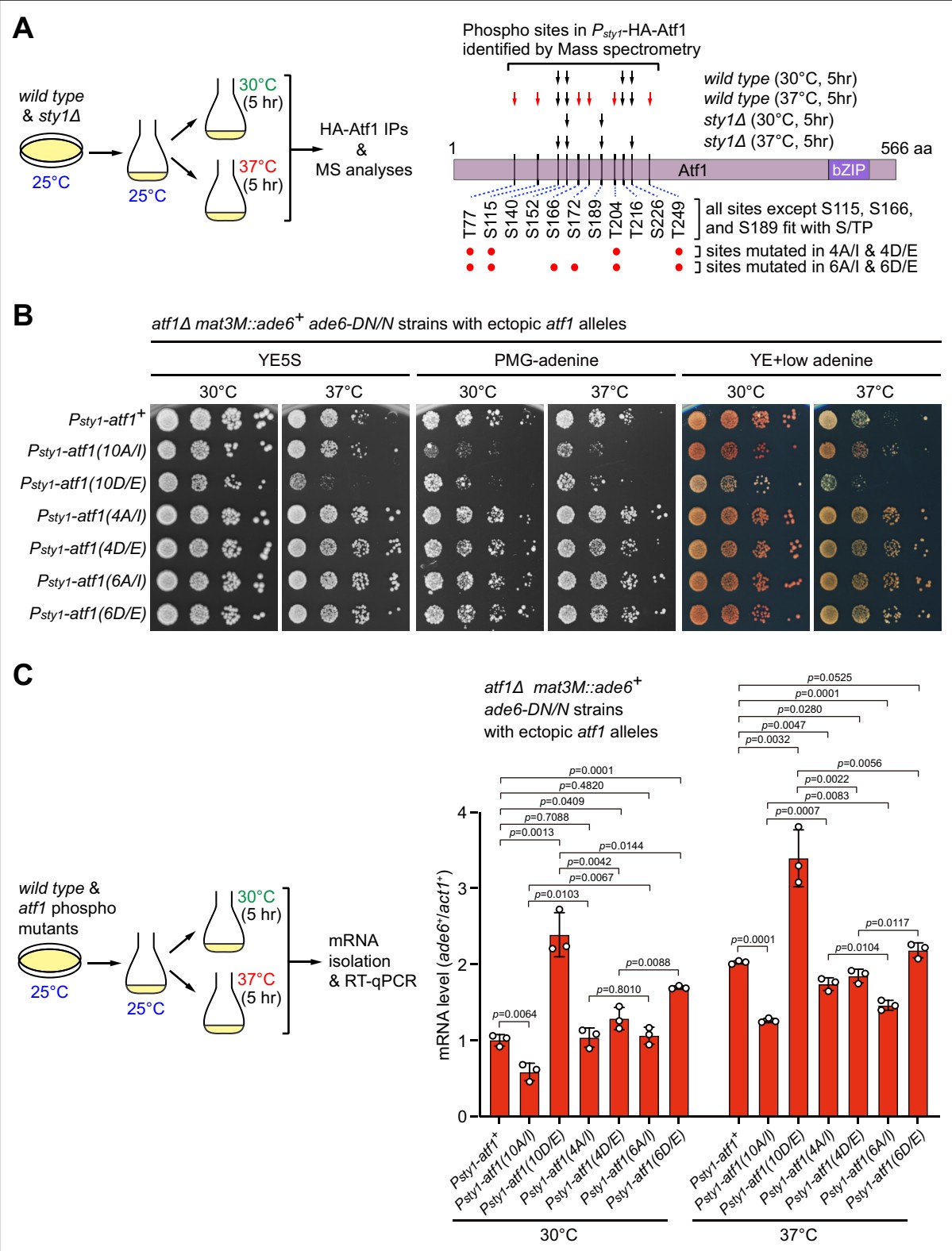

**Figure 6.** Identification of major Sty1-dependent phosphorylation sites in Atf1 upon heat stress. (**A**) (Left) Schematic depicting the experimental flow of the purification of HA-Atf1 for mass spectrometry (MS) identification of phosphorylation sites. (Right) Summary of MS-identified Atf1 phosphorylation sites in vivo in $P_{sty1}$-HA-atf1 cells. Arrows indicate detected phosphorylated residues and red arrows denote sites specifically enriched in wild type cells grown at 37°C. (**B**) Expression of the mat3M::ade6[+] reporter monitored by serial dilution spot assay in atf1Δ cells expressing Atf1 phospho mutants

*Figure 6 continued on next page*

*Figure 6 continued*

under $P_{sty1}$ promoter. (**C**) RT-qPCR analyses of the *mat3M::ade6⁺* reporter in *atf1Δ* cells expressing Atf1 phospho mutants under $P_{sty1}$ promoter. (Left) Schematic depicting the experimental flow of culturing and mRNA extraction.

The online version of this article includes the following source data and figure supplement(s) for figure 6:

**Source data 1.** Raw data of RT-qPCR.

**Figure supplement 1.** Identification of Atf1 residues phosphorylated by Sty1 in vivo.

**Figure supplement 2.** Mass spectrometry (MS) spectra from mass spectrometric analyses of Atf1.

**Figure supplement 3.** Phosphorylation of Ser438 in Atf1 is not involved in heat stress-induced defective heterochromatic maintenance at the mating-type region.

**Figure supplement 3—source data 1.** Raw data of RT-qPCR.

assumed that phosphorylation of Ser2 and Ser4 likely contributes to negative regulation of heterochromatin maintenance at *mat* locus, based on our observations in Atf1(10A/I) or Atf1(10D/E) mutants, which include mutations at both Ser2 and Ser4 (see data above). For Ser438, it has never been individually mutated to test its effect. To investigate whether the Ser438 of Atf1 was also involved in heterochromatin maintenance at the *mat* locus, we constructed yeast strains expressing Atf1-S348A, Atf1-S348D, Atf1(11A/I) (i.e. 10A/I plus S348A) or Atf1(11D/E) (i.e. 10A/I plus S348D) mutants and examined their effect on expression of the *mat3M::ade6⁺* reporter. We found that Atf1-S348A alone could not rescue heat stress-induced defective reporter silencing at the mating-type region, and when it was combined with Atf1(10A/I), it did not enhance the rescuing effect on heat stress-induced defective reporter silencing (**Figure 6—figure supplement 3**). These results indicated that phosphorylation of S348 site is not essential for regulating heterochromatin establishment and maintenance at the mating-type region.

Overall, these data suggested that Sty1 phosphorylates only some of the 11 putative MAPK phosphorylation sites in Atf1 upon heat stress, and the rest of the residues, such as Ser140, Ser152, and Ser226, are probably phosphorylated by other kinase(s).

## Tethering Swi6^HP1 to the *mat* locus rescues heat stress-induced defective heterochromatic maintenance at the *mat* locus

Our above results demonstrated that lowered Swi6^HP1 abundance at the *mat* locus under heat stress is very likely the major cause for defective heterochromatin stability. To further test this assumption, we adopted an artificial tethering system involving bacterial tetracycline operator sequence (*tetO*) and repressor protein (TetR^off) (**Bayne et al., 2010**; **Ragunathan et al., 2015**). A sequence containing four *tetO* upstream of *ade6⁺* reporter gene was inserted into *mat3M* locus (*mat3M::4xtetO-ade6⁺*), and Swi6 lacking its CD was fused with TetR^off (TetR^off-Swi6^ΔCD), which should allow the fusion protein to bind specifically at *4xtetO-ade6⁺* (**Figure 7A**). This indeed resulted in efficient recruitment of TetR^off-Swi6^ΔCD to the reporter at both 30°C and 37°C (**Figure 7B**). And cells expressing TetR^off-Swi6^ΔCD but not TetR^off or Swi6^ΔCD only rendered completely red colonies at 37°C (**Figure 7C**), indicating full rescue of silencing defects at the *mat* locus under heat stress. Our RT-qPCR and ChIP-qPCR analyses showed that the mRNA level of the *ade6⁺* was decreased dramatically and H3K9me3 level was restored, respectively, in cells expressing TetR^off-Swi6^ΔCD at 37°C (**Figure 7D**), demonstrating that tethering Swi6^HP1 to the *mat* locus was sufficient to rescue heat stress-induced defective epigenetic maintenance of heterochromatin. This also supports the idea that low Swi6^HP1 affinity and abundance at the *mat* locus brought about by MAPK-dependent phosphorylation of Atf1 is the major contribution factor for loss of heterochromatin at higher temperatures.

## Increased heterochromatin spreading in *epe1Δ* alleviates silencing defects at the *mat* locus upon heat stress

At the *mat* locus, both HDACs Sir2 and Clr3 contribute to heterochromatin spreading in concert with RNAi-directed heterochromatin nucleation (**Shankaranarayana et al., 2003**; **Yamada et al., 2005**). We noticed that Clr3 level was slightly decreased at regions flanking the *cenH* nucleation center but not at the *cenH* itself at high temperature (**Figure 4—figure supplement 1B**). In *S. pombe*, several factors have been identified to be required for preventing uncontrolled heterochromatin spreading

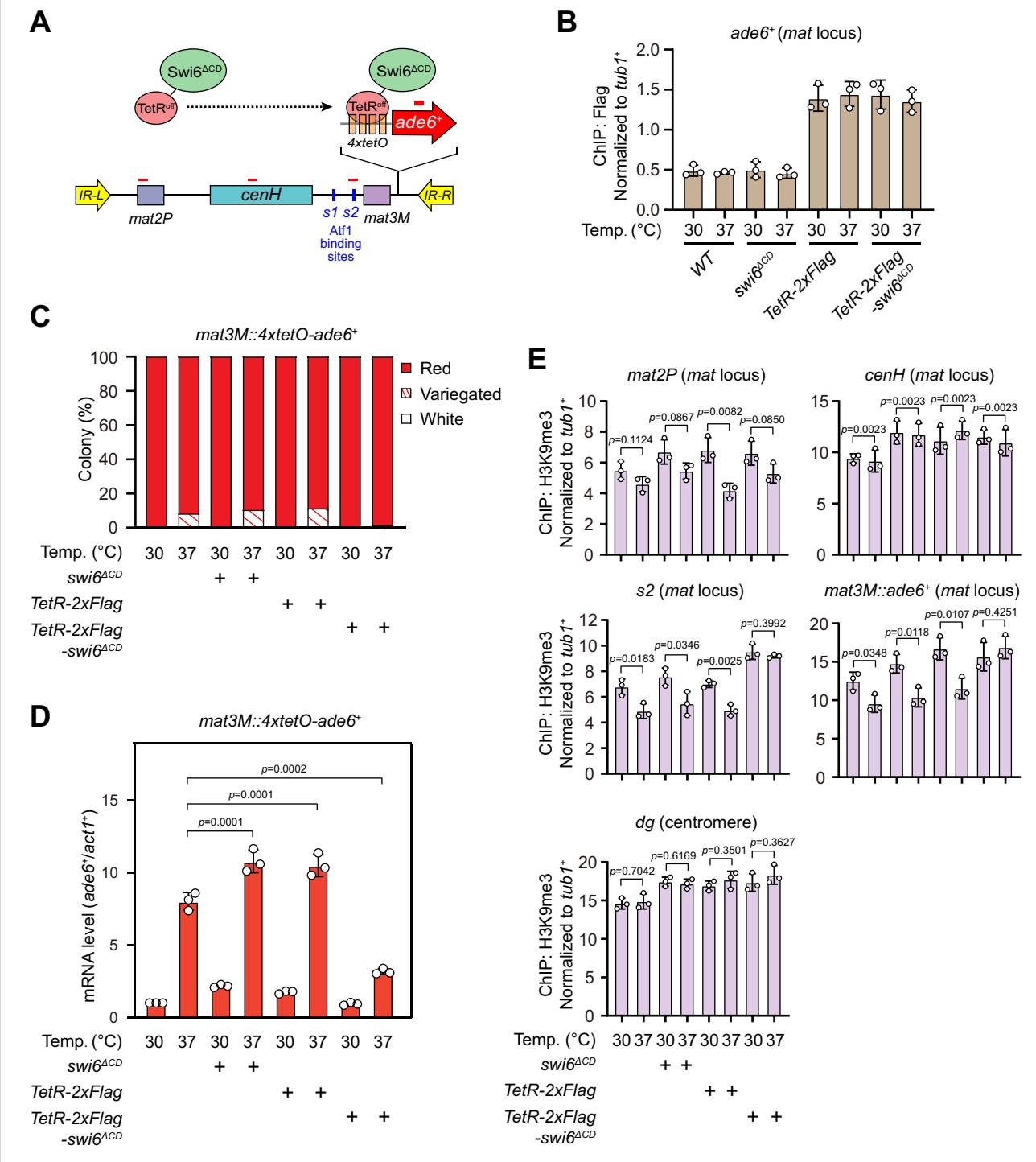

**Figure 7.** Tethering Swi6 [HP1] to the *mat3M*-flanking site rescues heat stress-induced defective epigenetic maintenance of heterochromatin at the *mat* locus. (**A**) Schematic of tethering Swi6[HP1] to the *mat* locus. A sequence containing four tetracycline operators located upstream of *ade6+* reporter gene (*4xtetO-ade6+*) was inserted next to *mat3M* locus, and Swi6 lacking CD domain was fused with TetR[off] (TetR[off]-Swi6[ΔCD]) and a 2xFlag tag. Primer positions for RT-qPCR or chromatin immunoprecipitation (ChIP) analysis are indicated (red bars). (**B**) ChIP-qPCR analyses of Flag-tagged TetR[off]-Swi6[ΔCD] at *4xtetO-ade6+* locus. Relative enrichment of TetR[off]-Swi6[ΔCD] was normalized to that of a *tub1+* fragment. Error bars represent standard deviation of three experiments. (**C**) Expression of the *mat3M::4xtetO-ade6+* reporter monitored by colony color assay. n>500 colonies counted for each sample. (**D**) RT-qPCR analyses of the *mat3M::4xtetO-ade6+* reporter. (**E**) ChIP-qPCR analyses of H3K9me3 levels at heterochromatic loci in Swi6[HP1]-tethered cells.

The online version of this article includes the following source data for figure 7:

**Source data 1.** Raw data of colony color assay, RT-qPCR, TetR-Flag/H3K9me3 ChIP.

and massive ectopic heterochromatin, notably the JmjC domain-containing protein Epe1 (*Braun et al., 2011*; *Wang et al., 2013*; *Zofall and Grewal, 2006*), the histone acetyltransferase Mst2 (*Wang et al., 2015*), and the transcription elongation complex Paf1C, which includes five subunits Paf1, Leo1, Cdc73, Prf1, and Tpr1 (*Kowalik et al., 2015*; *Sadeghi et al., 2015*). If the heterochromatin defects at the *mat* locus under heat stress are also due to the compromised spreading mediated by Clr3, then we would expect that a genetic background that is more permissive to heterochromatin spreading might overcome this barrier and rescue silencing defects. Intriguingly, we found that *epe1Δ*, but not *mst2Δ* or *leo1Δ*, indeed moderately rescued the silencing defects at the *mat* locus based on our silencing assays and RT-qPCR analyses of the *mat3M::ade6+* transcripts (*Figure 8A and B*). Moreover, ChIP-qPCR analysis also showed that H3K9me3 level at the *mat* locus in *epe1Δ* cells was more robust than wild type, *mst2Δ* or *leo1Δ* cells at 37°C (*Figure 8C*).

Based on above data, we were suspicious that more Epe1 might enrich at the *mat* locus under heat stress to antagonize heterochromatic silencing. To test this possibility, we measured Epe1 abundance at the *mat* locus under heat stress by ChIP-qPCR. In contrast to our anticipation, the levels of Epe1 at the *mat* locus were similar in cells cultured at 30°C and 37°C (*Figure 8D*), indicating Epe1 is not actively involved in competition with Swi6<sup>HP1</sup> at this site.

## Discussion

Using fission yeast as the model organism, previous studies have shown that heterochromatic silencing at pericentromeres is largely stable under chronic heat stress conditions (*Oberti et al., 2015*), but heterochromatin becomes unstable and a significant de-repression occurs at the mating-type region at elevated temperatures (*Greenstein et al., 2018*; *Nickels et al., 2022*), although both loci are within the major constitutive heterochromatin regions. It has been established that the temperature insensitivity of centromeric heterochromatin is mainly attributed to the buffering effect of the protein disaggregase Hsp104, which actively prevents formation of Dicer aggregations in cytoplasmic inclusions and promotes the recycling of resolubilized Dcr1 (*Oberti et al., 2015*). However, the mechanistic details and the physiological relevance of the loss of heterochromatin stability at the *mat* locus under similar environmental stress are unknown.

### Instability of ATF/CREB family protein-dependent temperature-sensitive heterochromatin may operate through distinct mechanisms in eukaryotes

In the current study, we revisited the unusual temperature-sensitive heterochromatin at the mating-type region in fission yeast. It is known that at this locus, both RNAi machinery and multiple factors, including transcription factors Atf1/Pcr1 and Deb1 and the ORC, are required for local heterochromatin formation (*Greenstein et al., 2018*; *Jia et al., 2004a*; *Kim et al., 2004*; *Nickels et al., 2022*; *Thon et al., 1999*; *Wang et al., 2021*; *Yamada et al., 2005*). Among these factors, Atf1/Pcr1 heterodimer binds to the *cis* elements flanking the *mat3M* cassette, which are mainly composed of closely juxtaposed 137 bp DNA sequence containing *s1* and *REIII* elements (*Jia et al., 2004a*; *Kim et al., 2004*; *Nickels et al., 2022*; *Thon et al., 1999*; *Wang et al., 2021*; *Yamada et al., 2005*). We found that heat stress does not drive the release of Atf1 from heterochromatin (*Figure 3A*), instead the binding affinity between Atf1 and heterochromatin protein Swi6<sup>HP1</sup> is severely compromised (*Figures 4A and 9*). This is distinct from the cases in *Drosophila*, mouse and swine, where the release of phosphorylated Atf1 homologs dATF-2 or ATF7 from heterochromatin is the major cause of the disrupted heterochromatin at elevated temperatures (*Liu et al., 2019*; *Seong et al., 2011*; *Sun et al., 2023*).

On the other hand, similarity does exist between fission yeast and higher eukaryotes (such as *Drosophila* and mammals) in that phosphorylation of ATF/CREB family proteins by MAPK are involved in promoting heterochromatin instability. We provided a few lines of evidence to support the idea that phosphorylation of Atf1 by MAPK in *S. pombe* is directly responsible for defective heterochromatin assembly at the *mat* locus under heat stress. First, non-phosphorylatable Atf1 (i.e. Atf1(10A/I)) is indeed more abundantly loaded at both the mating-type region and other Atf1 targets at euchromatic loci than wild type Atf1 (*Figure 3F*), which efficiently maintains the enrichment of H3K9me3 and Swi6<sup>HP1</sup> and its binding affinity with Swi6<sup>HP1</sup> (*Figures 3G and 4A*). Second, much reduced binding between phosphomimetic Atf1 (i.e. Atf1(10D/E)) and Swi6<sup>HP1</sup> can be similarly detected between wild

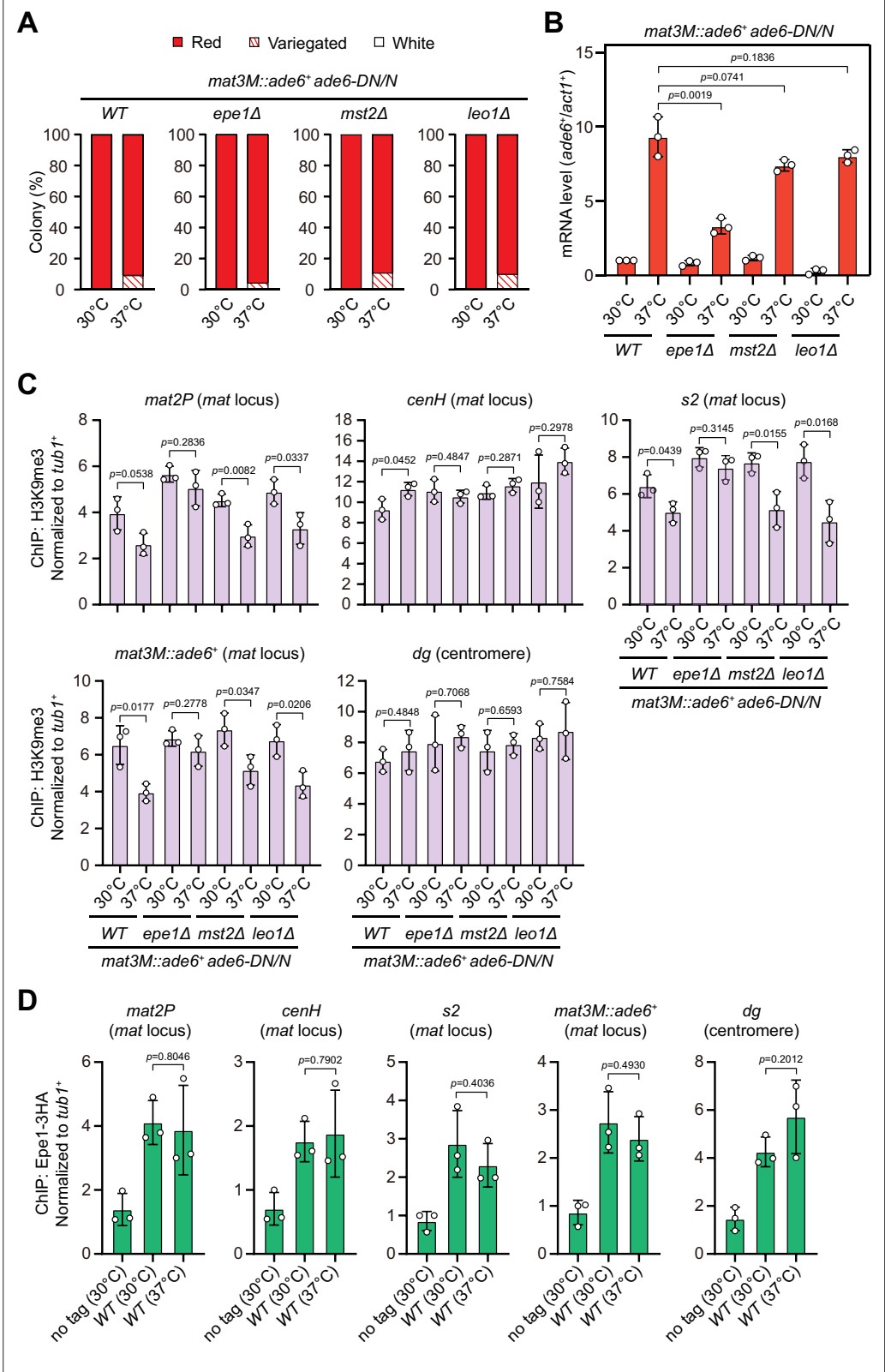

**Figure 8.** Deletion of anti-silencing factor Epe1 rescues heat stress-induced defective epigenetic maintenance of heterochromatin at mating-type region. (**A**) Expression of the *mat3M::ade6⁺* reporter monitored by colony color assay in *epe1Δ*, *mst2Δ*, or *leo1Δ* cells. n>500 colonies counted for each sample. (**B**) RT-qPCR analyses of the *mat3M::ade6⁺* reporter in *epe1Δ*, *mst2Δ*, or *leo1Δ* cells. (**C**) ChIP-qPCR analyses of H3K9me3 levels at

*Figure 8 continued on next page*

*Figure 8 continued*

heterochromatic loci in *epe1Δ, mst2Δ*, or *leo1Δ* cells. (**D**) ChIP-qPCR analyses of Epe1 levels at heterochromatic loci in wild type cells grown at 30°C or 37°C. Relative enrichment of Epe1-3HA was normalized to that of a *tub1*[+] fragment. Error bars represent standard deviation of three experiments. Two-tailed unpaired *t*-test was used to derive p-values.

The online version of this article includes the following source data for figure 8:

**Source data 1.** Raw data of colony color assay, RT-qPCR, H3K9me3/Epe1-3HA ChIP.

type Atf1 and Swi6[HP1] when MAPK is constitutively activated (i.e. in *wis1-DD* mutants) (**Figures 4A and 5E**). It still remains mysterious how MAPK-mediated phosphorylation of Atf1 loses its affinity toward Swi6[HP1], which requires more detailed study in the future.

It is noteworthy that our mass spectrometry analyses on Atf1 purified from wild type and *sty1Δ* cells grown at 37°C pinpointed at least six residues of Atf1 as heat-induced and Sty1-dependent phosphorylation sites (**Figure 6A, Figure 6—figure supplement 1**, and **Figure 6—figure supplement 2**). Interestingly, these sites largely fall in the middle region of Atf1, this is similar to those 11 putative MAPK phosphorylation sites which were originally identified purely based on their fit with ST/P motif. This is also consistent with previous observation that the central domain of Atf1 harboring six Ser/Thr sites affects heterochromatin assembly capacity of Atf1 at the *mat* locus (**Fraile et al., 2022**), reinforcing the idea that phosphorylation of multiple sites is required for negative regulatory effect of Atf1 on heterochromatin maintenance at *mat* locus upon heat stress. We also noticed that a few among those 11 putative MAPK phosphorylation sites (such as Ser2, Ser4, and Ser438) which fit with ST/P motif in Atf1 did not show up in our mass spectrometric identifications, most likely due to technical limitations of spectrometry analyses. At least our genetic analyses on Ser438 mutants excluded its involvement in influencing gene silencing at the *mat* locus (**Figure 6—figure supplement 3**). In addition, we also identified three residues Ser140, Ser152, and Ser226 as heat-induced but Sty1-independent phosphorylation sites, indicating that other kinase(s) may also collaborate with MAPK to affect heterochromatin maintenance at *mat* locus upon heat stress.

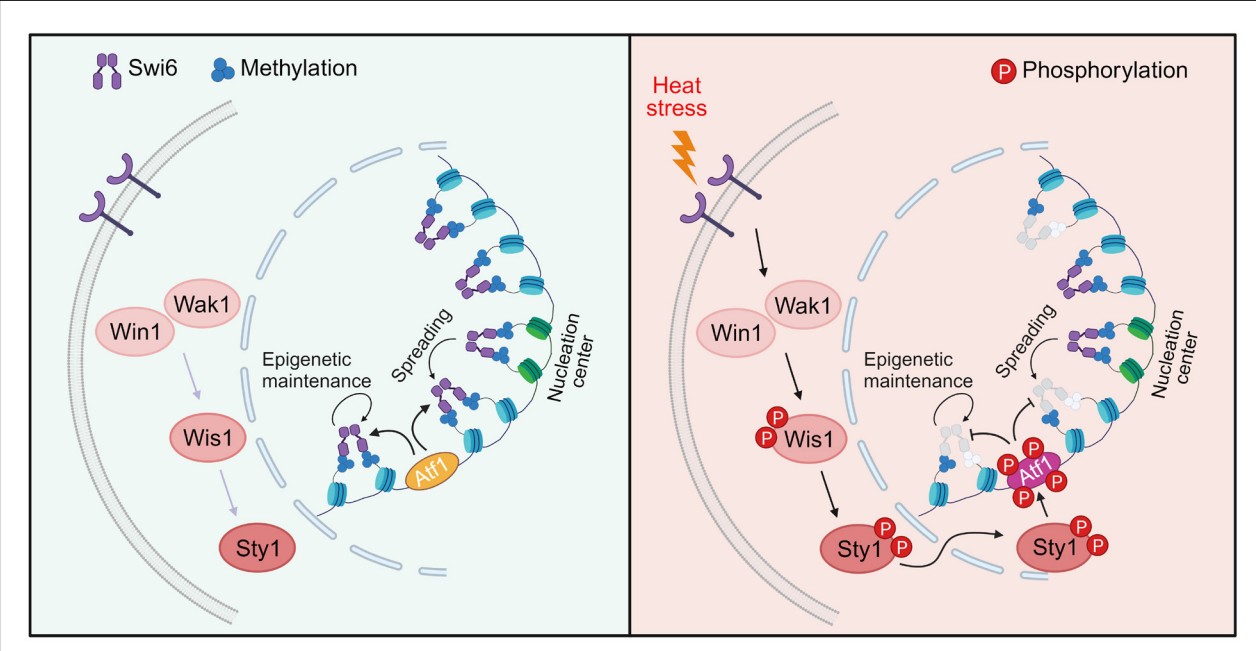

**Figure 9.** Proposed model for how heat-induced and mitogen-activated protein kinase (MAPK)-dependent Atf1 phosphorylation provokes epigenetic changes at the *mat* locus in fission yeast. Atf1 plays a dominating role in heterochromatin spreading and integrity maintenance at *mat* locus at normal temperature, but MAPK-mediated Atf1 phosphorylation compromises its binding affinity to Swi6[HP1], therefore attenuates heterochromatin stability under heat stress.

In previous studies performed in higher eukaryotes, heterochromatin stability has been mostly characterized based on the enrichment of ATF/CREB family protein itself, heterochromatin protein HP1 and H3K9me within examined heterochromatin sites/regions under heat stress. In our current study, in addition to altered binding between Atf1 and Swi6[HP1] under heat stress, we also found that Clr3, one of the histone modifying enzymes, fails to be efficiently recruited to the *cenH*-flanking sites within the fission yeast *mat* locus (*Figure 4—figure supplement 1B*), and *epe1Δ* is able to alleviate silencing defects at this locus (*Figure 8A–C*). Although we did not observe the weakened binding between Atf1 and Clr3 by in vitro pull-down assay (*Figure 4—figure supplement 1A*), or enhanced binding of Epe1 at *cenH*-flanking sites (*Figure 8D*), it does not exclude the possibility that Clr3-recruiting activity of Atf1 is only compromised or Epe1 is more retained respectively in a sub-population of heterochromatic nucleosomes, which cannot be detected by our current methods. Alternatively, heat stress-induced decreased recruitment of Clr3 at *mat* locus could be attributed to compromised binding between Swi6 and Atf1, because Clr3 interacts with Swi6 and Swi6 is involved in Clr3 spreading throughout the 20 kb heterochromatic *cenH*-flanking domain as previously reported (*Yamada et al., 2005*). Furthermore, our observation that H3K9me3 enrichment is reduced within *cenH* element-surrounding regions but not at *cenH* site itself (*Figure 1E*) can also be explained by the fact that Clr3 contributes to heterochromatin spreading and maintenance at those regions by stabilizing H3K9 trimethylation (*Yamada et al., 2005*), but heterochromatin nucleation at *cenH* site mainly requires RNAi-directed mechanism. Thus, it is fairly possible that loss of heterochromatin stability at high temperature may also involve histone-modifying enzymes to curb heterochromatin formation in higher eukaryotes.

## Physiological relevance of the loss of heterochromatin stability at *mat* locus in fission yeast

In mammals, the loss of the ATF/CREB family protein-dependent heterochromatin maintenance may cause detrimental consequences. It has been shown that p38-dependent phosphorylation of ATF7 in mice causes its release from the promoters of genes encoding either TERRA (telomere repeat containing RNA) in the sub-telomeric region or Cdk inhibitor p16[Ink4a], which disrupts heterochromatin and induces TERRA or cellular senescence and a shorter lifespan, respectively (*Liu et al., 2019*; *Maekawa et al., 2019*). TERRA can be transgenerationally transmitted to zygotes via sperm and causes telomere shortening in the offspring (*Liu et al., 2019*). During early porcine embryonic development, high temperatures also trigger the increased expression level of p38, which leads to ATF7 phosphorylation, heterochromatin disruption, and eventually the failure of blastocyst formation (*Sun et al., 2023*).

In fission yeast, the tightly silent mating-type region contributes to the mating-type switching to ensure the presence of almost equal number of cells with opposite mating types (*Klar, 2007*). When exposed to poor nitrogen source conditions, opposite mating type cells mate to form diploid zygotes, and undergo meiosis to form a zygote and ascus containing four spores subsequently (*Ohtsuka et al., 2022*). Previous studies have shown that the integrity of heterochromatin at the *mat* locus is crucial for efficient mating-type switching in fission yeast (*Hansen et al., 2011*; *Jia et al., 2004b*). Indeed, fission yeast cells spend much effort to establish and maintain stable heterochromatin at the *mat* locus by employing multiple mechanisms and factors (*Hansen et al., 2011*; *Jia et al., 2004a*; *Jia et al., 2004b*; *Kim et al., 2004*; *Thon et al., 1999*; *Wang et al., 2021*; *Yamada et al., 2005*). One very recent study has proposed that the transcription factor Atf1 not only involves in heterochromatin maintenance at the *mat* locus, but also acts in parallel with RNAi machinery and multiple histone-modifying enzymes during heterochromatin establishment steps (*Nickels et al., 2022*). Even so, fission yeast still suffers the stress-induced defective heterochromatic maintenance at this important locus in its genome. The presence of Atf1 is unable to efficiently fend off phenotypic variation, while its functional disruption can cause even much severe silencing defects at 37°C, as demonstrated by cells with *s1* and *REIII* elements (i.e. the major Atf1 binding sites) deleted (*Nickels et al., 2022*).

It is generally believed that yeast spores have higher stress tolerance than vegetative cells, which should be helpful for them to survive the unfavorable environment before they meet more friendly conditions. However, it is quite anti-intuitional that fission yeast cells do not mate and therefore fail to undergo meiosis and sporulation at temperatures above 33°C (*Brown et al., 2020*). The heat stress-induced heterochromatic disruption might interfere with the mating-type switching and reduce

mating efficiency, but it seems to be innocuous to vegetatively growing cells. For fission yeast, this feature may be regarded as an 'intrinsic flaw' which prevents its use of a better strategy to 'escape' from stress and has not been fixed during its evolution. How much heterochromatic maintenance defects at the *mat* locus induced by heat stress is contributing to this feature will be an interesting question for future studies to solve.

## Materials and methods
### Yeast methods

*S. pombe* strains and DNA oligos used in this study are listed in *Supplementary file 1*. Yeast strains with C-terminal tagged or deletion of genes were generated by a PCR-based module method with the DNA sequence information obtained from PomBase (https://www.pombase.org). Genetic crosses and general yeast techniques were performed as described previously (*Forsburg and Rhind, 2006*; *Moreno et al., 1991*). Liquid cultures or solid agar plates consisting of rich medium (YE5S) or minimal medium (PMG5S) containing 4 g/L sodium glutamate as a nitrogen source with appropriate supplements were used as described previously (*Forsburg and Rhind, 2006*; *Moreno et al., 1991*). G418 disulfate (Sigma-Aldrich; A1720), hygromycin B (Sangon Biotech; A600230), or nourseothiricin (clonNAT; Werner BioAgents; CAS#96736-11-7) was used at a final concentration of 100 µg/mL where appropriate. 5-FOA (Shanghai Nuotai Chemical Co. Ltd, Shanghai, China) was added in solid YE5S or PMG5S plates to get a final concentration of 0.15% for counterselection of *ura4+*. *sty1-T97A* was inhibited with 5 µM or 10 µM 3-BrB-PP1 (Abcam; ab143756) added in liquid or solid media respectively.

To construct yeast strains carrying genomically integrated $P_{sty1}$-*HA-atf1+* and $P_{sty1}$-*HA-atf1 mutants* at the ectopic locus *leu1+*, the DNA fragment of $P_{sty1}$::*HA-atf1* was first amplified from a yeast strain carrying $P_{sty1}$::*HA-atf1::leu1+* (a kind gift from Elena Hidalgo) and cloned into the pJK148-based vector. Then, mutations of Atf1 phosphorylation sites from Ser or Thr to Ala, Ile, Glu, or Asp were introduced via Quikgene method (*Mao et al., 2011*), this generated a series of vectors of pJK148-$P_{sty1}$::*HA-atf1* with *atf1* mutant derivatives. Finally, the resultant plasmids were linearized by *Nru*I and integrated at the *leu1-32* locus, generating a series of strains of *leu1-32::$P_{sty1}$::HA-atf1(WT)::leu1+* and *leu1-32::$P_{sty1}$::HA-atf1(mutant derivatives)::leu1+*.

To construct the strains containing *atf1* mutants at the endogenous locus, *atf1Δ::ura4+* cells were transformed with pBluescript-*atf1* mutant constructs containing 5′ and 3′ noncoding flanks. The integration candidates were selected based on their resistance to 5-FOA and integration of the *atf1* mutations was verified by PCR followed by DNA sequencing.

To construct the strain containing *4xtetO-ade6+* reporter at the *mat* locus, the vector pBW5/6-4xTetO-ade6+ (a kind gift from Robin C Allshire) was digested with *Pst*I and inserted into the *ura4+* locus in strains with *mat3M(Eco*RV)::*ura4+*. To construct the plasmid pHBKA81-TetR^off^-2xFlag-swi6^ΔCD^, TetR^off^-2xFlag was amplified from a vector pDUAL-TetR^off^-2xFlag-Stc1 (a kind gift from Robin C Allshire) and cloned into upstream of *swi6+* in the plasmid pHBKA81-swi6-hyg^R^ to generate pHBKA81-TetR^off^-2xFlag-swi6. CD (80–133aa) of Swi6 was then deleted by Quikgene method (*Mao et al., 2011*). Finally, the resultant plasmid pHBKA81-TetR^off^-2xFlag-swi6^ΔCD^ was linearized by *Apa*I and integrated into the *lys1+* locus, generating the strain *lys1Δ::$P_{adh81}$-TetR^off^-2xFlag::hyg^R^*.

Introduction of *leu1-32::$P_{sty1}$::HA-atf1(WT)::leu1+*, *atf1* mutants at the endogenous locus, and *lys1Δ::$P_{adh81}$-TetR^off^-2xFlag::hyg^R^* into other genetic backgrounds was accomplished using standard *S. pombe* mating, sporulation, and tetrad dissection techniques.

### Reporter gene silencing assay

The *ura4+* silencing was assessed by growth on PMG5S without uracil or with 1 mg/mL 5-FOA, which is toxic to cells expressing *ura4+*. *ade6+* silencing was assessed by growth on YE5S with 75 mg/L or 0.5 mg/L adenine, the latter was referred to as YE5S with low adenine. For serial dilution assays, three serial 10-fold dilutions were made, and 5 µL of each was spotted on plates with the starting cell number of $10^4$.

For *ade6+* gene silencing recovery assays, about 500 cells of each strain were plated on YE5S with low adenine medium and incubated at 30°C or 37°C, the variegated colonies were counted manually. Three variegated colonies grown on solid YE5S with low adenine at 37°C were picked

and re-plated on YE5S with low adenine medium and then incubated at 30°C or 37°C to assess the silencing recovery rate.

## Protein extraction and immunoblotting

For total protein extraction, 20 OD$_{600}$ units of *S. pombe* cells at mid-log phase were collected, followed by lysing with glass bead disruption using Bioprep-24 homogenizer (ALLSHENG Instruments, Hangzhou, China) in 200 µL lysis buffer containing urea (0.12 M Tris-HCl, pH 6.8, 20% glycerol, 4% SDS, 8 M urea, 0.6 M β-mercaptoethanol).

Western blotting was performed essentially as previously described (*Wang et al., 2012*). The primary antibodies used for immunoblot analysis of cell lysates were mouse monoclonal anti-Atf1 (abcam, ab18123) (1:2000) and rabbit monoclonal phospho-p38 MAPK (Cell Signaling, 4511T) (1:1000). Cdc2 was detected using rabbit polyclonal anti-PSTAIRE (Santa Cruz Biotechnology, sc-53) (1:10,000 dilution) as loading controls. Secondary antibodies used were goat anti-mouse or goat anti-rabbit polyclonal IgG (H+L) HRP conjugates (Thermo Fisher Scientific; #31430 or #32460) (1:5000–10,000).

## Phosphorylation site identification by mass spectrometry

HA-Atf1 was purified from 5 L cultures of wild type or *sty1Δ* cells carrying *P$_{sty1}$-HA-atf1* grown at 30°C or 37°C for 5 hr after being cultured at 25°C. Cells were disrupted and cell lysates were prepared as described above for routine immunoblotting experiments. Proteins were immunoprecipitated by anti-HA magnetic beads (MedChemExpress; HY-K0201).

For mass spectrometry analyses, purified samples were first run on PAGE gels, after staining of gels with Coomassie blue, excised gel segments were subjected to in-gel trypsin (Promega, V5111) digestion and dried. Samples were then analyzed on a nanoElute (plug-in V1.1.0.27; Bruker, Bremen, Germany) coupled to a timsTOF Pro (Bruker, Bremen, Germany) equipped with a CaptiveSpray source. Peptides were separated on a 15 cm × 75 µm analytical column, 1.6 µm C18 beads with a packed emitter tip (IonOpticks, Australia). The column temperature was maintained at 55°C using an integrated column oven (Sonation GmbH, Germany). The column was equilibrated using 4 column volumes before loading sample in 100% buffer A (99.9% MilliQ water, 0.1% FA) (both steps performed at 980 bar). Samples were separated at 400 nL/min using a linear gradient from 2% to 25% buffer B (99.9% ACN, 0.1% FA) over 90 min before ramping to 37% buffer B (10 min), ramp to 80% buffer B (10 min) and sustained for 10 min (total separation method time 120 min). The timsTOF Pro (Bruker, Bremen, Germany) was operated in PASEF mode using Compass Hystar 6.0. Mass range 100–1700 m/z, 1/K0 start 0.6 V·s/cm$^2$ end 1.6 V·s/cm$^2$, Ramp time 110.1 ms, Lock duty cycle to 100%, Capillary voltage 1600 V, Dry gas 3 L/min, Dry temp 180°C, PASEF settings: 10 MS/MS scans (total cycle time 1.27 s), Charge range 0–5, Active exclusion for 0.4 min, Scheduling target intensity 10,000, Intensity threshold 2500, CID collision energy 42 eV. All raw files were analyzed by PEAKS Studio Xpro software (Bioinformatics Solutions Inc, Waterloo, ON, Canada). Data was searched against the *S. pombe* proteome sequence database (Uniprot database with 5117 entries of protein sequences at https://www.uniprot.org/proteomes/UP000002485). De novo sequencing of peptides, database search, and characterizing specific PTMs were used to analyze the raw data; false discovery rate was set to ≤1%, and [–10*log(p)] was calculated accordingly where p is the probability that an observed match is a random event. The PEAKS used the following parameters: (i) precursor ion mass tolerance, 20 ppm; (ii) fragment ion mass tolerance, 0.05 Da (the error tolerance); (iii) tryptic enzyme specificity with two missed cleavages allowed; (iv) monoisotopic precursor mass and fragment ion mass; (v) a fixed modification of cysteine carbamidomethylation; and (vi) variable modifications including N-acetylation of proteins and oxidation of Met.

## In vitro pull-down assay

All recombinant bacterially produced His-Swi6, GST-Clr3, GST-Clr4, and MBP-Clr6 were expressed in *Escherichia coli* BL21 (DE3) cells and purified on Ni$^+$ Sepharose 6 Fast Flow (for 6His fusions; GE Healthcare, 17531806), Glutathione Sepharose 4B (for GST fusions; GE Healthcare, 17075601), or amylose resin (for MBP fusions; New England BioLabs, E8021) according to the manufacturer's instructions. Yeast cells were lysed by glass bead disruption using Bioprep-24 homogenizer (ALLSHENG Instruments) in NP40 lysis buffer (6 mM Na$_2$HPO$_4$, 4 mM NaH$_2$PO$_4$, 1% NP-40, 150 mM NaCl, 2 mM EDTA, 50 mM NaF, 0.1 mM Na$_3$VO$_4$) plus protease inhibitors. Each purified recombinant protein (about 1 µg)

immobilized on resins/beads was incubated with cleared yeast cell lysates for 1–3 hr at 4°C. Resins/beads were thoroughly washed and suspended in SDS sample buffer, and then subject to SDS-PAGE electrophoresis and Coomassie brilliant blue staining. Western blotting was performed to detect the association between Swi6, Clr3, Clr4, and Clr6 and yeast-expressed Atf1 and Pcr1.

## RT-qPCR analysis

Total RNA was extracted using the TriPure Isolation Reagent (Roche). Reverse transcription and quantitative real-time PCR were performed with PrimeScript RT Master Mix (Takara, RR037A) and TB Green Premix Ex Taq II (Takara, RR820A) in a StepOne real-time PCR system (Applied Biosystems). The relative mRNA level of the target genes in each sample was normalized to $act1^+$.

## ChIP-qPCR analysis

The standard procedures of chromatin immunoprecipitation were used as previously described (*Cam and Whitehall, 2016*) with slight modifications. In brief, cells grown in YE5S at 30°C or 37°C to mid-log phase were crosslinked with 3% paraformaldehyde for 30 min at room temperature (at 18°C for Swi6). Thirty $OD_{600}$ units of *S. pombe* cells were collected and lysed by beads disruption using Bioprep-24 homogenizer (ALLSHENG Instruments) in 1 mL lysis buffer (50 mM HEPES-KOH pH 7.5, 140 mM NaCl, 1 mM EDTA, 1% Triton X-100, 0.1% deoxycholate) with protease inhibitors. Lysates were sonicated to generate DNA fragments with sizes of 0.5–1 kb and immunoprecipitated with mouse monoclonal anti-H3K9me2 (abcam, ab1220), rabbit polyclonal anti-H3K9me3 (abcam, ab8898), mouse monoclonal anti-Atf1 (abcam, ab18123), rabbit polyclonal anti-Swi6 (abcam, ab188276), goat polyclonal anti-myc (abcam, ab9132), or mouse monoclonal anti-Flag M2 (Sigma, F1804), and subjected to pull-down with rProteinA Sepharose Fast Flow (GE Healthcare, 17127901) or ProteinG Sepharose 4 Fast Flow (GE Healthcare, 17061801). The crosslinking was reversed, and DNA was purified with the ChIP DNA Clean and Concentrator (ZYMO RESEARCH, D5201) kit. Quantitative real-time PCR was performed with TB Green Premix Ex Taq II (Takara, RR820A) in a StepOne real-time PCR system (Applied Biosystems). For normalization, serial dilutions of DNA were used as templates to generate a standard curve amplification for each pair of primers, and the relative concentration of target sequence was calculated accordingly. The enrichment of a target sequence in immunoprecipitated DNA over whole-cell extract was calculated and normalized to that of a reference fragment $tub1^+$ as previously described (*Wang et al., 2013*).

## Statistical analysis

For quantitative analyses of RT-qPCRs and ChIP-qPCR, experiments were repeated three times, and the mean value and standard deviation (s.d.) for each sample was calculated. In order to determine statistical significance of each pair of data, two-tailed unpaired *t*-tests were performed and p-values were calculated using GraphPad Prism 7. $p < 0.05$ was considered statistically significant.

## Acknowledgements

We thank Robin C Allshire, Marc Buhler, Amikam Cohen, Elena Hidalgo, Genevieve Thon, Janni Petersen, and National BioResource Project (NBRP), Japan (http://yeast.nig.ac.jp/yeast/) for fission yeast strains or plasmids; Da-jie Deng for initial construction of some relevant yeast strains. We also thank Yaying Wu, Zheni Xu, and Chang-chuan Xie for mass spectrometry experiments and data analyses. This work was supported by grants from the National Natural Science Foundation of China (No. 32170731, No. 30871376) to QW Jin.

## Additional information

### Funding

| Funder | Grant reference number | Author |
|---|---|---|
| National Natural Science Foundation of China | 32170731 | Quan-wen Jin |

| Funder | Grant reference number | Author |
|---|---|---|
| National Natural Science Foundation of China | 30871376 | Quan-wen Jin |

The funders had no role in study design, data collection and interpretation, or the decision to submit the work for publication.

## Author contributions

Li Sun, Conceptualization, Data curation, Formal analysis, Investigation, Visualization, Methodology, Writing - original draft; Libo Liu, Chunlin Song, Conceptualization, Data curation, Formal analysis, Investigation, Visualization; Yamei Wang, Conceptualization, Formal analysis, Supervision, Investigation, Writing - review and editing; Quan-wen Jin, Conceptualization, Resources, Data curation, Formal analysis, Supervision, Funding acquisition, Investigation, Visualization, Writing - original draft, Project administration, Writing - review and editing

## Author ORCIDs

Li Sun http://orcid.org/0000-0003-0708-2881
Quan-wen Jin http://orcid.org/0000-0001-6146-6910

## Decision letter and Author response

Decision letter https://doi.org/10.7554/eLife.90525.sa1
Author response https://doi.org/10.7554/eLife.90525.sa2

# Additional files

## Supplementary files

- Supplementary file 1. Yeast strains and primers for qPCR used in this study. (a) Yeast strains used in this study. (b) Primers used for RT-qPCR and qPCR.
- MDAR checklist

## Data availability

The authors confirm that all data supporting the findings of this study are available within the manuscript main figures, figure supplements, source data files and Supplementary file 1. The mass spectrometry proteomics data have been deposited to the ProteomeXchange Consortium via the PRIDE partner repository with the dataset identifier PXD048330.

The following dataset was generated:

| Author(s) | Year | Dataset title | Dataset URL | Database and Identifier |
|---|---|---|---|---|
| Sun L, Liu L, Song C, Wang Y, Jin QW | 2024 | Sty1 kinase-dependent phosphosite mapping on Atf1 | https://www.ebi.ac.uk/pride/archive/projects/PXD048330 | PRIDE, PXD048330 |

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
