## [Editor Report]

This fundamental study demonstrates that MAPK signaling under heat stress phosphorylates ATF in fission yeast leading to de-repression of the mating type locus. The compelling work shows the phosphorylation of ATF reduces its interactions with the heterochromatin protein Swi6 causes its loss at the mating type locus. This is an example of how signaling regulates transcription factor:co-repressor interactions to control gene expression.

---

## [Decision Letter]

**Decision letter after peer review:**

Thank you for submitting your article "Heat Stress-induced Activation of MAPK Pathway Attenuates Atf1-dependent Epigenetic Inheritance of Heterochromatin in Fission Yeast" for consideration by *eLife*. Your article has been reviewed by 3 peer reviewers, one of whom is a member of our Board of Reviewing Editors, and the evaluation has been overseenKevin Struhl as the Senior Editor.

The reviewers, Senior and Reviewing Editors have discussed your manuscript and concluded that additional controls and experiments are necessary to solidify the advances in your manuscript over previously published work. In particular it was determined that analysis of individual phosphorylation site mutations under expression of the endogenous promoter is required to advance the work and confirm the role of MAPKs. This and other requested experiments are listed below.

Essential revisions:

1. Some main conclusions are based upon experiments in which Atf1 and Atf1 mutants are expressed constitutively under the control of the Psty1 promoter.

Atf1 levels in *S. pombe* cells depend on Sty1 activity in two ways. Firstly, an increased Sty1 dependent transcription and secondly increased stability via Sty1 dependent phosphorylation of the Atf1 protein. Both the transcriptional induction as well as phosphorylation of Atf1 during stress response is known to have a temporal pattern. So, the endogeneous levels of a phosphorylation defective mutant of Atf1 expressed from its native promoter are expected to be very different from that expressed constitutively in the experimental model that the authors have chosen. Specifically, the levels of Atf1 are expected to be much higher in the conditions chosen by the author and not representative of the levels that would be there in *S. pombe* cells during heat stress. This difference would have a significant effect on the interactions of Atf1 with the other factors. A higher Atf1 level would also exaggerate the influence of Sty1 dependent phosphorylation on the mechanism described. Hence it’s not very clear, to what extent the mechanisms described in this manuscript would be operational in *S. pombe* cells during heat stress. Appropriate experiments using phosphorylation mutants under the control of the endogeneous Atf1 promoter are therefore essential to validate the conclusions made by the authors.

2. Under normal temperature conditions, earlier studies have implicated that phosphorylation of Atf1 by Sty1 is important for heterochromatin initiation. At the same time a constitutively expressed Atf1.10M mutant (under the Psty1 promoter) has been shown to be able to establish silencing at the mat locus while an Atf1.10D mutant failed to do so (Fraile et al., iScience. 2022 Aug 19; 25(8): 104820). Reports also exist which indicate that kinase activity of Sty1 is important for heterochromatin formation at the mat locus. (Kim et al., J Biol Chem,2004 Oct 8;279(41):42850-9) The authors need to discuss the contrasting implications of the existing reports and how these results co-relate with their own conclusions.

3. Atf1 functions are known to be influenced strongly by Pcr1 both during stress responses and heterochromatin formation. In fact, Pcr1 is also known to interact with Swi6 (Kim et al., J Biol Chem,2004 Oct 8;279(41):42850-9). During Heat stress the transcriptional induction of pcr1+ is much more than that of Atf1^+^. Atf1 and Pcr1 are also known to influence each other's expression levels. The mechanistic details presented in this study therefore remain largely incomplete in the absence of an assessment of whether and how Pcr1 levels are affected in a cell expressing Atf1 mutants constitutively. The mechanistic details remain elusive in the absence of that.

4. Since phosphorylation of Atf1 is presented as the basis of the alteration in heterochromatin state at the mat locus, this study would have benefitted immensely by the further investigation of which one (or more) of the 10 indicated phosphorylatable sites plays the major role in the suggested mechanism. This could have been easily done via genetic approaches similar to using the Atf1.10A/I mutant or via biochemical approaches that exactly identify which residues in Atf1 remain phosphorylated under these conditions.

5. Studies with analog sensitive mutants of Sty1 would have been necessary to conclusively prove that it is the kinase activity of Sty1 which is responsible for the observed effects. While experiments with wis1-DD mutants do provide and indication, the observation needed further validation.

6. The only well characterized observations in the manuscript therefore are the ones that show the changes in recruitment of Swi6 and the clr proteins, which is important but in the absence of a conclusive mechanism that leads to these changes, the result remains incomplete.

7. Figure S2B: Why the ratio of GFP/Cdc2 and the mRNA level of gfp/act1 are inconsistent, especially for imr1R::GFP strain.

8. Figure 2C-D: What would happen to the expression of ade6, cenH and dg when cells were treated at 37°C?

9. Figure 3B: In addition to the 10 phosphorylation sites on the left end of Atf1, there is also a S348, why this site was not studied? Mutation of Atf1 (10A/I) at 37℃ did not reduce the expression of ade6 to wild-type level, which could be due to Atf1-S348 is still active. Or does it indicate that there are other pathways regulating gene silencing under heat stress?

10. Figure 3C: From the western blots, we can tell that the level of Atf1 (10A/I) was decreased and the Atf1 (10D/E) protein was recovered. Does phosphorylation of Atf1 affect Atf1 protein level?

11. It may be favorable to present the ade6 reporter data in Figure 1 first and follow it with discussion of the confirming ura4 data which is in supplemental data.

12. Further discussion is required to explain why upon heat stress K9me2 is down at all heterochromatic regions while H3K9me3 is only down at mat. Something is also happening at other heterochromatic sites. Is it similar but insufficient to relieve silencing at those loci or some different phenomenon.

13. Figure 4 presents compelling data that bacterially expressed Swi6 loses its binding to ATF in extracts made from heat-treated cells. Of course, the gold standard of such experiments is reciprocal IPs with endogenous proteins. The authors should perform such experiments or provide a compelling technical description of why it is not feasible in this case.

---

## [Author Response]

Essential revisions:1. Some main conclusions are based upon experiments in which Atf1 and Atf1 mutants are expressed constitutively under the control of the Psty1 promoter.Atf1 levels in *S. pombe* cells depend on Sty1 activity in two ways. Firstly, an increased Sty1 dependent transcription and secondly increased stability via Sty1 dependent phosphorylation of the Atf1 protein. Both the transcriptional induction as well as phosphorylation of Atf1 during stress response is known to have a temporal pattern. So, the endogeneous levels of a phosphorylation defective mutant of Atf1 expressed from its native promoter are expected to be very different from that expressed constitutively in the experimental model that the authors have chosen. Specifically, the levels of Atf1 are expected to be much higher in the conditions chosen by the author and not representative of the levels that would be there in *S. pombe* cells during heat stress. This difference would have a significant effect on the interactions of Atf1 with the other factors. A higher Atf1 level would also exaggerate the influence of Sty1 dependent phosphorylation on the mechanism described. Hence its not very clear, to what extent the mechanisms described in this manuscript would be operational in *S. pombe* cells during heat stress. Appropriate experiments using phosphorylation mutants under the control of the endogeneous Atf1 promoter are therefore essential to validate the conclusions made by the authors.

To follow your suggestion, we have constructed new strains with phosphorylation-defective mutant (Atf1(10A/I)) and phospho-mimicking mutant (Atf1(10D/E)) expressed under the control of the endogeneous *atf1* promoter (*P_atf1_*). We performed several experiments using these newly constructed strains:

1) By performing Western blotting, we compared the Atf1 protein levels of Atf1(10A/I) or Atf1(10D/E) expressed from its native promoter (*P_atf1_*) and from the constitutive *sty1* promoter (*P_sty1_*). Indeed, as expected, the levels of both wild type and phospho mutant of Atf1 are higher when they are expressed from *sty1* promoter than endogeneous *atf1* promoter-driven expression at both 30 ºC and 37 ºC.

2) By colony color assays and RT-qPCR analyses of the *mat3M::ade6^+^* reporter, we could demonstrate that phospho-defective version of Atf1 (i.e. *atf1(10A/I)*) is still able to significantly restore epigenetic silencing and phospho-mimicking mutant (Atf1(10D/E)) severely disrupts *mat3M::ade6^+^* silencing at the *mat* locus under heat stress. More strikingly, *P_atf1_*-Atf1(10D/E) also compromises epigenetic silencing at the *mat* locus even at 30 ºC.

3) By ChIP analyses, we also examined the recruitment of Atf1 at *s1* and *s2* sites, and found that the binding of Atf1 phospho mutants at the *mat* locus is not largely altered.

Overall, our new results demonstrated that although Atf1(10A/I) or Atf1(10D/E) expressed from its native promoter is less abundant than *P_sty1_*-driven Atf1 mutants, phospho-defective Atf1 is still able to maintain epigenetic silencing to certain degree and Atf1(10D/E) causes opposite effect on epigenetic silencing and heterochromatin stability at the *mat* locus. Thus, these results validated our observations in *P_sty1_*-driven Atf1 mutants that phosphorylation of Atf1 by MAPK imposes negative effect on heterochromatin initiation and maintenance upon heat stress.

We have presented these new results in Figure 3—figure supplement 2 in our revised manuscript.

2. Under normal temperature conditions, earlier studies have implicated that phosphorylation of Atf1 by Sty1 is important for heterochromatin initiation. At the same time a constitutively expressed Atf1.10M mutant (under the Psty1 promoter) has been shown to be able to establish silencing at the mat locus while an Atf1.10D mutant failed to do so (Fraile et al., iScience. 2022 Aug 19; 25(8): 104820). Reports also exist which indicate that kinase activity of Sty1 is important for heterochromatin formation at the mat locus. (Kim et al., J Biol Chem,2004 Oct 8;279(41):42850-9) The authors need to discuss the contrasting implications of the existing reports and how these results co-relate with their own conclusions.

Actually, both the two previous reports (Kim et al., J Biol Chem. 2004, PMID: 15292231; DOI: 10.1074/jbc.M407259200 and Fraile et al., iScience, 2022, PMID: 35992058; DOI: 10.1016/j.isci.2022.104820) you mentioned support the same, instead of contrasting, conclusion that phosphorylation of Atf1 by Sty1 imposes negative effect on heterochromatin initiation and maintenance, rather than promotes heterochromatin formation at the *mat* locus. In Kim et al. (2004) paper, it clearly demonstrated that the absence of *sty1* or *wis1* (i.e. in *sty1Δ* or *wis1Δ* mutants) enhances heterochromatin stability at the *mat* locus (please see Figure 3A, B in that paper). This observation is recapitulated in Fraile et al. (2022) by showing that Sty1 phosphorylation-deficient mutant Atf1.10M (equivalent to Atf1(10A/I)) is able to establish silencing at the *mat* locus.

Consistent with these two previous papers, our data demonstrated that constitutively activated MAPK by Wis1-DD mutant disrupted heterochromatin stability at the *mat* locus even under normal temperature conditions (25 ºC and 30 ºC) and removal of Sty1 kinase activity abated heterochromatic silencing defects at the *mat* locus. Please see our results presented in Figure 5B, Figure 5—figure supplement 1B and Figure 5—figure supplement 2.

3. Atf1 functions are known to be influenced strongly by Pcr1 both during stress responses and heterochromatin formation. In fact, Pcr1 is also known to interact with Swi6 (Kim et al., J Biol Chem,2004 Oct 8;279(41):42850-9). During Heat stress the transcriptional induction of pcr1+ is much more than that of Atf1^+^. Atf1 and Pcr1 are also known to influence each other's expression levels. The mechanistic details presented in this study therefore remain largely incomplete in the absence of an assessment of whether and how Pcr1 levels are affected in a cell expressing Atf1 mutants constitutively. The mechanistic details remain elusive in the absence of that.

We agree that we should have included results from investigation on the potential involvement of Pcr1 in heterochromatin formation and maintenance during heat stress. Following your suggestion, we have performed Western blotting to examine the Pcr1 protein levels in *atf1Δ*, *P_sty1_-atf1(WT)*, *P_sty1_-atf1(10A/I)*, and *P_sty1_-atf1(10D/E)* cells. We have also employed in vitro binding assays and ChIP analyses to assess whether the interaction between Pcr1 and Swi6 and the enrichment of Pcr1 at the *mat* locus are affected by Atf1(10A/I) and Atf1(10D/E) mutants, respectively.

We found that, not like Atf1, the binding of Pcr1 to Swi6^HP1^ was not affected by either heat stress or phosphorylation status of Atf1. In addition, absence of *atf1* and Atf1 phospho mutants did not alter enrichment of Pcr1 at the *mat* locus, though *P_sty1_*-*HA-atf1(10A/I)* elevated protein abundance of Pcr1 at both 30 ºC and 37 ºC.

These new data suggested that Atf1 and Pcr1 respond differently and separately to heat stress in heterochromatin maintenance at the *mat* locus, and the disrupted Swi6^HP1^-Atf1 interaction is the major cause of compromised heterochromatin inheritance at high temperature.

We have presented these new results in Figure 4—figure supplement 2.

4. Since phosphorylation of Atf1 is presented as the basis of the alteration in heterochromatin state at the mat locus, this study would have benefitted immensely by the further investigation of which one ( or more) of the 10 indicated phosphorylatable sites plays the major role in the suggested mechanism. This could have been easily done via genetic approaches similar to using the Atf1.10A/I mutant or via biochemical approaches that exactly identify which residues in Atf1 remain phosphorylated under these conditions.

We agree with you that it is possible that only one or a few among 10 phosphorylatable sites within the N-terminal half of Atf1 plays the major role in regulating epigenetic maintenance of heterochromatin at the mating-type region. To further narrow down the phosphorylation sites, we purified Atf1 from both wild type and *sty1Δ* cells incubated at 30 ºC or 37 ºC and performed Mass Spectrometry analyses to exactly identify residues in Atf1 with Sty1-dependent phosphorylation under heat stress. We have pinpointed at least 6 residues (T77, S115, S166, S172, T204 and T249) of Atf1 as heat-induced and Sty1-dependent phosphorylation sites. Interestingly, these sites largely fall in the middle region of Atf1, this is similar to those 11 putative MAPK phosphorylation sites which were originally identified purely based on their fit with ST/P motif.

We mutated all 6 identified sites to either non-phosphorylatable alanines (Ala, A) and isoleucines (Ile, I) (i.e. Atf1(6A/I)), or phosphomimetic aspartic acids (Asp, D) and glutamic acids (Glu, E) (i.e. Atf1(6D/E)) and observed decreased or increased silencing and mRNA levels of *mat3M::ade6^+^* reporter respectively, although the effect was not as strong as in Atf1(10A/I) or Atf1(10D/E) mutants. In contrast, mutating 4 out of 6 these sites led to only modest effect on expression of *mat3M::ade6^+^* at high temperature, reinforcing the idea that phosphorylation of multiple but not only a few sites is required for negative regulatory effect of Atf1 on heterochromatin maintenance at *mat* locus upon heat stress.

We have presented our new data in Figure 6A, Figure 6—figure supplement 1 and Figure 6—figure supplement 2.

5. Studies with analog sensitive mutants of Sty1 would have been necessary to conclusively prove that it is the kinase activity of Sty1 which is responsible for the observed effects. While experiments with wis1-DD mutants do provide and indication, the observation needed further validation.

Following your suggestion, in order to examine whether removal of Sty1 kinase activity could relieve the negative effect of *wis1-DD* mutants on heterochromatin stability at the *mat* locus, we have introduced either *sty1* deletion mutant (*sty1∆*) or ATP analogue-sensitive mutant of *sty1* (*sty1-T97A*) into *wis1-DD* mutant background.

We have performed a series of experiments using these newly constructed strains carrying *sty1* mutations, including:

1) Serial dilution drop test to follow expression of the *K*∆*::ade6^+^* reporter;

2) RT-qPCR analyses to follow mRNA levels of the *K*∆*::ade6^+^* reporter;

3) Western blotting analyses of the phosphorylated Sty1 and the total protein of Atf1;

4) In vitro pull-down assays to examine the binding affinity between Atf1 and Swi6^HP1^;

5) ChIP-qPCR analyses of H3K9me3 and Swi6 levels at heterochromatic loci.

Our new results showed that the kinase activity of Sty1 is indeed responsible for the defective epigenetic maintenance of heterochromatin at the mating-type region we observed in *wis1-DD* mutants, in which MAPK signaling pathway is constitutively activated. Thus, these results validated our former conclusion that constitutive activation of MAPK signaling and resulted Atf1 phosphorylation by Sty1 can eventually lead to defective epigenetic maintenance and inheritance of heterochromatin at the *mat* locus even under normal temperatures.

We have now incorporated these new results in revised Figure 5 and Figure 5—figure supplement 2, and moved the data presented in former Figure 5 to Figure 5—figure supplement 1.

6. The only well characterized observations in the manuscript therefore are the ones that show the changes in recruitment of Swi6 and the clr proteins, which is important but in the absence of a conclusive mechanism that leads to these changes, the result remains incomplete.

During revision of our manuscript, we have performed extra experiments, including:

1) Colony color assays, serial dilution drop test and RT-qPCR analyses of the mat3M::ade6+ reporter in atf1(10A/I) and atf1(10D/E) mutants under the control of the endogeneous atf1 promoter (Figure 3—figure supplement 2);

2) Western blotting analyses of Pcr1 protein levels in Psty1-atf1(10A/I) and Psty1-atf1(10D/E) mutants (Figure 4—figure supplement 2);

3) ChIP analyses and in vitro binding assays to assess whether the enrichment of Pcr1 at the mat locus and the interaction between Pcr1 and Swi6 is affected by Atf1(10A/I) and Atf1(10D/E) mutants (Figure 4—figure supplement 2);

4) A series of experiments to examine whether removal of Sty1 kinase activity by introducing either sty1 deletion mutant (sty1∆) or ATP analogue-sensitive mutant of sty1 (sty1-T97A) into wis1-DD mutant background could relieve the negative effect of constitutive activation of MAPK Sty1 on kΔ::ade6+ reporter gene silencing, binding affinity between Atf1 and Swi6HP1 and heterochromatin stability at the mat locus (Figure 5 and Figure 5—figure supplement 2).

5) We have also tried to narrow down the Sty1-dependent phosphorylation sites in Atf1 at high temperature by Mass spectrometry mapping and analyses of individual phosphorylation site mutations.

Basically, our new results validated the major conclusion from our first version manuscript that phosphorylation of Atf1 by Sty1 imposes negative effect on heterochromatin spreading and maintenance upon heat stress. Our new results also revealed a mechanism that the compromised binding affinity between Swi6^HP1^ and Atf1, but not Pcr1, is the major factor involved in heterochromatin spreading and maintenance during heat stress.

Now, we think our revised manuscript has been substantially strengthened by the addition of these new data, therefore represents a much better account of our work.

7. Figure S2B: Why the ratio of GFP/Cdc2 and the mRNA level of gfp/act1 are inconsistent, especially for imr1R::GFP strain.

Earlier versions of wild-type GFP has been found to be thermosensitive, namely the maturation or stability is compromised for GFP or GFP-fusion proteins expressed in vertebrate or human cells and in yeast cells when incubated at 37 ºC (Please see several earlier publications, such as: Kaether and Gerdes, FEBS Lett, 1995 (PMID: 7649270/DOI: 10.1016/0014-5793(95)00765-2); Ogawa et al., PNAS, 1995 (PMID: 8524871/DOI: 10.1073/pnas.92.25.11899); Lim et al., J Biochem, 1995 (PMID: 8537302/ DOI: 10.1093/oxfordjournals.jbchem.a124868); Siemering et al., Curr Biol. 1996 (PMID: 8994830/DOI: 10.1016/s0960-9822(02)70789-6)). We have added citations of two of these publications in our revised manuscript as supporting evidence of our observation (please see page 6, line #171).

The inconsistency between GFP/Cdc2 and the mRNA level of *gfp*/*act1* can be explained by the thermosensitivity of GFP at 37 ºC for both *imr1R::GFP* and *mat3M::GFP* strains. It is noteworthy that the de-repression of the *mat3M::GFP* reporter at 37 ºC judged by its protein level is clearly detectable and significant (Figure 1—figure supplement 2B, left panel), even though it is known that GFP is unstable at 37 ºC. We assume that if the high temperature-insensitive fluorescence proteins, such as EGFP (enhanced GFP) or mGFP (monomer GFP), were used as the reporter at *mat3M*, the detected de-repression based on protein products should have been much more striking at 37 ºC.

8. Figure 2C-D: What would happen to the expression of ade6, cenH and dg when cells were treated at 37°C?

During revision of our manuscript, we have repeated RT-qPCR analyses of the *mat3M::ade6^+^* reporter, *cenH* and *dg* in *dcr1∆* strain background with red or variegated colonies, and confirmed that the expression of *ade6^+^* and *cenH* is inhibited in red colonies and relieved in variegated colonies, and *dg* transcription is de-repressed in cells from both red colonies and variegated colonies. This is true for cells grown at both 30 ºC and 37 ºC.

We have added these new data in Figure 2—figure supplement 1.

9. Figure 3B: In addition to the 10 phosphorylation sites on the left end of Atf1, there is also a S348, why this site was not studied? Mutation of Atf1 (10A/I) at 37℃ did not reduce the expression of ade6 to wild-type level, which could be due to Atf1-S348 is still active. Or does it indicate that there are other pathways regulating gene silencing under heat stress?

One very recent study has shown that the constitutively expressed Atf1(10A/I) mutant (under the *sty1^+^* promoter, *P_sty1_*) is able to promote heterochromatin establishment and maintenance at the *mat* locus under normal temperature conditions (Fraile *et al.*, iScience. 2022, PMID: 35992058; DOI: 10.1016/j.isci.2022.104820). Consistently, our data showed that expression of *P_sty1_*-*atf1(10A/I)* also rescues the heat stress-induced defective heterochromatic maintenance at the *mat* locus (see our data shown in Figure 3D, E and G). These findings raised the possibility that removal of phosphorylation of 10 out of the 11 serine or threonine on the left end of Atf1 is sufficient for heterochromatin establishment and maintenance at the mating-type region.

Nevertheless, to satisfy your curiosity, we constructed yeast strains expressing Atf1-S348A, Atf1-S348D, Atf1(11A/I) (i.e. 10A/I plus S348A) or Atf1(11D/E) (i.e. 10D/E plus S348D) mutants and examined their effect on expression of the *mat3M::ade6^+^* reporter. We found that Atf1-S348A alone could not rescue heat stress-induced defective reporter silencing at the mating-type region, and when it was combined with Atf1(10A/I), it did not enhance the rescuing effect on heat stress-induced defective reporter silencing. These results indicated that phosphorylation of S348 site is not essential for heterochromatin establishment and maintenance at the mating-type region. We have added these new data in Figure 6—figure supplement 3.

We have also tried to narrow down the Sty1-dependent phosphorylation sites in Atf1 at high temperature by Mass spectrometry mapping and analyzed silencing of *mat3M::ade6^+^* in individual phosphorylation site mutants. We have pinpointed at least 6 residues (T77, S115, S166, S172, T204 and T249) of Atf1 as heat-induced and Sty1-dependent phosphorylation sites. Interestingly, these sites largely fall in the middle region of Atf1, this is similar to those 11 putative MAPK phosphorylation sites which were originally identified purely based on their fit with ST/P motif. We have added these new data in Figure 6 and Figure 6—figure supplement 1.

For the issue that mutation of Atf1(10A/I) at 37℃ did not reduce the expression of *mat3M::ade6^+^* reporter to wild-type level, we agree that there may be additional phosphorylation sites within Atf1 which remain to be identified, or alternatively, other kinase(s) from additional pathways might also phosphorylate Atf1 to regulate gene silencing under heat stress.

However, it could also be due to several other reasons. First, although replacing Ser or Thr by Ala, Val or Ile is a very commonly used strategy to create the non-phosphorylatable mutation, it is actually quite “artificial” and is not necessarily always equivalent to nonphosphorylated state of those residues in some cases. Many examples of failed nonphosphorylation-mimicking strategy can be found throughout published literatures, one example has been reported in Koyano et al., Genes to Cells (2015) (PMID: 26525038; DOI: 10.1111/gtc.12309). The same issue also applies for phosphomimetic mutations by Glu or Asp, which do not necessarily result in expected phosphomimetic phenotypes. One of such examples was reported by Ruggiero et al., Current Biology (2020) (PMID: 31928870, DOI: 10.1016/j.cub.2019.11.054) in which the *mad3-S268E* mutant does not behave as phosphomimetic and does not delay mitotic slippage. For Atf1, perhaps a dynamic equilibrium and balance between phosphorylated and nonphosphorylated Atf1 is critical for fine-tuning Atf1 function in gene silencing maintenance at the mating-type region, and Atf1(10A/I) mutation could block its dynamic balance.

10. Figure 3C: From the western blots, we can tell that the level of Atf1 (10A/I) was decreased and the Atf1 (10D/E) protein was recovered. Does phosphorylation of Atf1 affect Atf1 protein level?

Actually, whether phosphorylation of Atf1 affects its protein level is still an unresolved issue.

Two earlier studies showed that a mutant version of Atf1, lacking all eleven putative mitogen-activated protein kinase (MAPK) sites (i.e. Atf1-11A/I), is less stable than its wild-type counterpart and accumulates to a lesser extent upon stress (Please see Lawrence *et al.* 2007, PMID: 17182615/DOI: 10.1074/jbc.M608526200; and Lawrence *et al.* 2009, PMID: 19836238/DOI: 10.1016/j.cub.2009.09.044). The instability of Atf1(11A/I) can be rescued by the absence of F-box protein Fbh1 or disruption of Atf1 interaction with Fbh1 (Lawrence *et al.* 2009, PMID: 19836238/DOI: 10.1016/j.cub.2009.09.044). In these studies, anti-Atf1 antibodies were used to detect wild type and phosphorylation mutants of Atf1. Based on these observations, it has been proposed that phosphorylation of Atf1 by Sty1 inhibits its ubiquitinylation-dependent degradation, and thus promotes its stabilization and accumulation, especially under stress.

However, one recent study challenged the above notion. Salat-Canela *et al.* noticed that protein level of HA-Atf1(10D/E) is significantly enhanced and that of HA-Atf1(10A/I) is decreased under stress when they used anti-Atf1 antibodies in Western blotting (Salat-Canela *et al.* 2017, PMID: 28652406; DOI: 10.1074/jbc.M117.794339), which is consistent with previous studies (Lawrence *et al.* 2007; Lawrence *et al.* 2009). Astonishingly, when they used anti-HA antibodies to detect HA-Atf1, HA-Atf1(10A/I) and HA-Atf1(10D/E) before and after H2O2 treatment, they found that the protein levels of wild type and phosphorylation mutants of Atf1 are comparable, demonstrating that the levels of HA-Atf1 are not enhanced upon stress (Salat-Canela *et al.* 2017, PMID: 28652406; DOI: 10.1074/jbc.M117.794339).

Because we used anti-Atf1 antibodies in our Western blotting experiments, so it is not surprising to detect decreased level of Atf1(10A/I) and stabilized Atf1(10D/E) under heat stress.

11. It may be favorable to present the ade6 reporter data in Figure 1 first and follow it with discussion of the confirming ura4 data which is in supplemental data.

Following your suggestion, we have re-organized the order of description for our presented data in Figure 1 and Figure 1—figure supplement 1. Now, it should represent a more logic flow of our data.

12. Further discussion is required to explain why upon heat stress K9me2 is down at all heterochromatic regions while H3K9me3 is only down at mat. Something is also happening at other heterochromatic sites. Is it similar but insufficient to relieve silencing at those loci or some different phenomenon.

As we have discussed in our manuscript, H3K9me3, but not H3K9me2, is a more reliable hallmark for heterochromatin, which has been documented recently by several studies, such as Jih et al., 2017 (PMID: 28682306, DOI: 10.1038/nature23267) and Cutter DiPiazza et al., 2021 (PMID: 34035174, DOI: 10.1073/pnas.2100699118). Thus, our observation that H3K9me3 enrichment is reduced within *mat* locus under heat stress is a good indication that heterochromatin maintenance is disrupted.

We agree that it indeed seems puzzling that H3K9me3 enrichment is reduced within *cenH* element-surrounding regions but not at *cenH* site itself (as we showed in our Figure 1E). We think this data can be explained by two facts about Clr3 function at *mat* locus revealed by one previous study (Yamada et al., 2005; PMID: 16246721, DOI: 10.1016/j.molcel.2005.10.002):

1) Clr3 interacts with Swi6 and Swi6 is involved in Clr3 spreading throughout the 20 kb heterochromatic *mat* locus;

2) Clr3 contributes to heterochromatin spreading and maintenance at *mat* locus by stabilizing H3K9 trimethylation (i.e. H3K9me3). The heterochromatin spreading and maintenance at *mat* locus is largely dependent on Clr3, but heterochromatin nucleation at *cenH* site mainly requires RNAi-directed mechanism.

Therefore, it is possible that heterochromatin spreading and maintenance at *cenH* element-surrounding regions are more sensitive to efficient Clr3 recruitment than the heterochromatin initiation at the *cenH* site.

We have added a brief relevant discussion in our revised manuscript (Please see page 20, line 549-558).

13. Figure 4 presents compelling data that bacterially expressed Swi6 loses its binding to ATF in extracts made from heat-treated cells. Of course, the gold standard of such experiments is reciprocal IPs with endogenous proteins. The authors should perform such experiments or provide a compelling technical description of why it is not feasible in this case.

We agree that the best way to examine the binding affinity of Swi6 to Atf1 or Pcr1 is performing reciprocal IPs with endogenous proteins, but practically it is not possible for us to detect the interaction between these proteins. We have tried co-IP experiments several times without success. The reason might be that only a small subpopulation of these proteins interact with each other in vivo and the amount of co-immunoprecipitated proteins is far below the detection limit by Western blotting, although Swi6, Atf1 and Pcr1 are among relatively abundant proteins in fission yeast.

Actually, although two previous studies successfully detected the interaction between Atf1 or Pcr1 and Swi6, neither of them employed co-IPs. Instead, one study used GST-pull down assay of overexpressed GST-Atf1 or overexpressed GST-Swi6 (Kim, H.S., et al., J Biol Chem. 2004, PMID: 15292231; DOI: 10.1074/jbc.M407259200), and the other study used in vitro translated and ^35^S-labeled Atf1 and Pcr1 and bacterially expressed GST-Swi6 (Jia S., et al., Science, 2004, PMID: 15218150; DOI: 10.1126/science.1099035). Although neither of these papers mentioned why they chose these “non-gold standard” techniques to deal with the protein interaction detection, we believe these two studies probably also encountered the similar technical difficulty as us in detecting the interaction between Atf1 or Pcr1 and Swi6 directly through co-IP. So, in this aspect, it is very likely that we are not alone in choosing alternative approach to tackle this technical difficulty in this specific case.